# GRESNET: GRAPH RESIDUAL NETWORK FOR REVIVING DEEP GNNS FROM SUSPENDED ANIMATION

## ABSTRACT

The existing graph neural networks (GNNs) based on the spectral graph convolutional operator have been criticized for its performance degradation, which is especially common for the models with deep architectures. In this paper, we further identify the suspended animation problem with the existing GNNs. Such a problem happens when the model depth reaches the suspended animation limit, and the model will not respond to the training data any more and become not learnable. Analysis about the causes of the suspended animation problem with existing GNNs will be provided in this paper, whereas several other peripheral factors that will impact the problem will be reported as well. To resolve the problem, we introduce the GRESNET (Graph Residual Network) framework in this paper, which creates extensively connected highways to involve nodes' raw features or intermediate representations throughout the graph for all the model layers. Different from the other learning settings, the extensive connections in the graph data will render the existing simple residual learning methods fail to work. We prove the effectiveness of the introduced new graph residual terms from the norm preservation perspective, which will help avoid dramatic changes to the node's representations between sequential layers. Detailed studies about the GRESNET framework for many existing GNNs, including GCN, GAT and LOOPYNET, will be reported in the paper with extensive empirical experiments on real-world benchmark datasets.

## 1 INTRODUCTION

Graph neural networks (GNN), e.g., *graph convolutional network* (GCN) Kipf & Welling (2016) and *graph attention network* (GAT) Veličković et al. (2018), based on the approximated spectral graph convolutional operator Hammond et al. (2011), can learn the representations of the graph data effectively. Meanwhile, such GNNs have also received lots of criticism, since as these GNNs' architectures go deep, the models' performance will get degraded, which is similar to observations on other deep models (e.g., convolutional neural network) as reported in He et al. (2015). Meanwhile, different from the existing deep models, when the GNN model depth reaches a certain limit (e.g., depth $\geq 5$ for GCN with the bias term disabled or depth $\geq 8$ for GCN with the bias term enabled on the Cora dataset), the model will not respond to the training data any more and become not learnable. Formally, we name such an observation as the GNNs' *suspended animation problem*, whereas the corresponding model depth is named as the *suspended animation limit* of GNNs. Here, we need to add a remark: to simplify the presentations in this paper, we will first take vanilla GCN as the base model example to illustrate our discoveries and proposed solutions in the method sections. Meanwhile, empirical tests on several other existing GNNs, e.g., GAT Veličković et al. (2018) and LOOPYNET Zhang et al. (2018), will also be studied in the experiment section of this paper.

As illustrated in Figure 1, we provide the learning performance of the GCN model on the Cora dataset, where the learning settings (including train/validation/test sets partition, algorithm implementation and fine-tuned hyper-parameters) are identical to those introduced in Kipf & Welling (2016). The GCN model with the bias term disable of seven different depths, i.e., GCN(1-layer)-GCN(7-layer), are compared. Here, the layer number denotes the sum of hidden and output layers, which is also equal to the number of spectral graph convolutional layers involved in the model. For instance, besides the input layer, GCN(7-layer) has 6 hidden layer and 1 output layer, both of which involve the spectral graph convolutional operations. According to the plots, GCN(2-layer) and GCN(3-layer) have comparable performance, which both outperform GCN(1-layer). Meanwhile, as the model depth increases from 3 to 7, its learning performance on both the training set

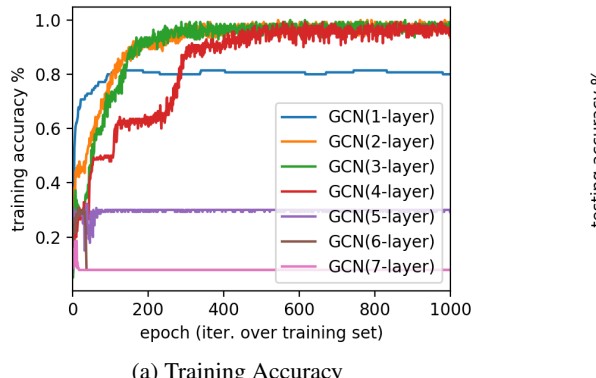 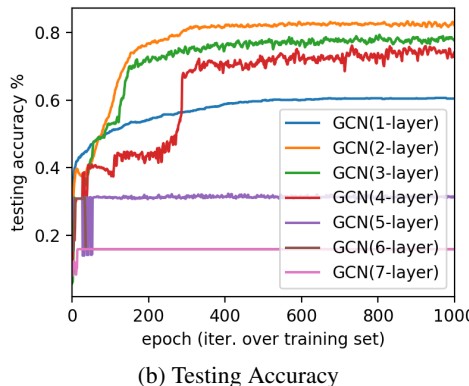

(a) Training Accuracy          (b) Testing Accuracy

Figure 1: The learning performance of GCN (bias disabled) with 1-layer, 2-layer, . . . , 7-layer on the Cora dataset. The x axis denotes the iterations over the whole training set. The y axis of the left plot denotes the training accuracy, and that of the right plot denotes the testing accuracy.

and the testing set degrades greatly. It is easy to identify that such degradation is not caused by over-fitting the training data. What's more, much more surprisingly, as the model depth goes deeper to 5 or more, it will suffer from the *suspended animation problem* and does not respond to the training data anymore. (Similar phenomena can be observed for GCN (bias enabled) and GAT as illustrated by Figures 9 and 10 in the appendix of this paper, whose suspended animation limits are 8 and 5, respectively. Meanwhile, on LOOPYNET, we didn't observe such a problem as shown in Figure 11 in the appendix, and we will state the reasons in Section 6 in detail.)

In this paper, we will investigate the causes of the GNNs' *suspended animation problem*, and analyze if such a problem also exists in all other GNN models or not. GNNs are very different from the traditional deep learning models, since the extensive connections among the nodes render their learning process no longer independent but strongly correlated. Therefore, the existing solutions proposed to resolve such problems, e.g., residual learning methods used in ResNet for CNN He et al. (2015), cannot work well for GNNs actually. In this paper, several different novel *graph residual terms* will be studied for GNNs specially. Equipped with the new *graph residual terms*, we will further introduce a new graph neural network architecture, namely *graph residual neural network* (GRESNET), to resolve the observed problem. Instead of merely stacking the spectral graph convolution layers on each other, the extensively connected high-ways created in GRESNET allow the raw features or intermediate representations of the nodes to be fed into each layer of the model. We will study the effectiveness of the GRESNET architecture and those different *graph residuals* for several existing vanilla GNNs. In addition, theoretic analyses on GRESNET will be provided in this paper as well to demonstrate its effectiveness from the norm-preservation perspective.

The remaining parts of this paper are organized as follows. In Section 2, we will introduce the related work of this paper. The suspended animation problem with the spectral graph convolutional operator will be discussed in Section 3, and the suspended animation limit will be analyzed in Section 4. Graph residual learning will be introduced in Section 5, whose effectiveness will be tested in Section 6. Finally, we will conclude this paper in Section 7.

## 2 RELATED WORK

**Graph Neural Network**: Graph neural networks Monti et al. (2017); Atwood & Towsley (2016); Masci et al. (2015); Kipf & Welling (2016); Battaglia et al. (2018); Bai et al. (2018); Scarselli et al. (2009); Zhou et al. (2018); Niepert et al. (2016) have become a popular research topic in recent years. Traditional deep models cannot be directly applied to graph data due to the graph interconnected structures. Many efforts have been devoted to extend deep neural networks on graphs for representation learning. GCN proposed in Kipf & Welling (2016) feeds the generalized spectral features into the convolutional layer for representation learning. Similar to GCN, deep loopy graph neural network Zhang (2018) proposes to update the node states in a synchronous manner, and it introduces a spanning tree based learning algorithm for training the model. LOOPYNET accepts nodes' raw features into each layer of the model, and it can effectively fight against the suspended animation problem according to the studied in this paper. GAT Veličković et al. (2018) leverages masked self-attentional layers to address the shortcomings of GCN. In this year, we have also witnessed some preliminary works on heterogeneous graph neural networks Wang et al. (2019); Liu et al. (2018). Similar to GCN, GEM Liu et al. (2018) utilizes one single layer of attention to capture the impacts of both neighbors and network heterogeneity, which cannot work well on real-world complex networks. Based on GAT, HAN Wang et al. (2019) learns the attention coefficients between

the neighbors based on a set of manually crafted meta paths Sun et al. (2011), which may require heavy human involvements. DIFNN Zhang et al. (2018) introduce a diffusive neural network for the graph structured data specifically, which doesn't suffer from the oversmoothing problem due to the involvement of the neural gates and residual inputs for all the layers. Due to the limited space, we can only name a few number of the representative graph neural network here. The readers are also suggested to refer to page[1], which provides a summary of the latest graph neural network research papers with code on the node classification problem.

**Residual Network**: Residual learning Srivastava et al. (2015); He et al. (2015); Bae et al. (2016); Han et al. (2016); Gomez et al. (2017); Tai et al. (2017); Yu et al. (2017); Ahn et al. (2018); Li et al. (2018a); Behrmann et al. (2019) has been utilized to improve the learning performance, especially for the neural network models with very deep architectures. To ease gradient-based training of very deep networks, Srivastava et al. (2015) introduces the highway network to allow unimpeded information flow across several layers. Innovated by the high-way structure, He et al. (2015) introduces the residual network to simplify highway network by removing the fusion gates. After that, residual learning has been widely adopted for deep model training and optimization. Bae et al. (2016) introduces residual learning for image restoration via persistent homology-guided manifold simplification; Tai et al. (2017); Li et al. (2018a) introduces a recursive residual network for image resolution adjustment. Some improvement of residual network has also been proposed in recent years. A reversible residual network is introduced in Gomez et al. (2017), where each layer's activations can be reconstructed exactly from the next layer's; Han et al. (2016) improves the conventional model shape with a pyramidal residual network instead; Yu et al. (2017) introduce the dilated residual network to increases the resolution of output feature maps without reducing the receptive field of individual neurons; and Ahn et al. (2018) studies the cascading residual network as an accurate and lightweight deep network for image super-resolution. Readers can also refer to He (2016) for a detailed tutorial on residual learning and applications in neural network studies.

# 3   SUSPENDED ANIMATION PROBLEM WITH GCN MODEL

In this part, we will provide an analysis about the *suspended animation* problem of the spectral graph convolutional operator used in GCN to interpret the causes of the observations illustrated in Figure 1. In addition, given an input network data, we will provide the theoretic bound of the *suspended animation limit* for the GCN model.

## 3.1   VANILLA GRAPH CONVOLUTIONAL NETWORK REVISIT

To make this paper self-contained, we will provide a brief revisit of the vanilla GCN model in this part. Formally, given an input network $G = (\mathcal{V}, \mathcal{E})$, its network structure information can be denoted as an adjacency matrix $\mathbf{A} = \{0, 1\}^{n \times n}$ (where $|\mathcal{V}| = n$). GCN defines the normalized adjacency matrix of the input network as $\hat{\mathbf{A}} = \tilde{\mathbf{D}}^{-\frac{1}{2}} \tilde{\mathbf{A}} \tilde{\mathbf{D}}^{-\frac{1}{2}}$, where $\tilde{\mathbf{A}} = \mathbf{A} + \mathbf{I}^{n \times n}$ and $\tilde{\mathbf{D}}$ is the diagonal matrix of $\tilde{\mathbf{A}}$ with entry $\tilde{\mathbf{D}}(i, i) = \sum_j \tilde{\mathbf{A}}(i, j)$. Given all the nodes in $\mathcal{V}$ together with their raw feature inputs $\mathbf{X} \in \mathbb{R}^{n \times d_x}$ ($d_x$ denotes the node raw feature length), GCN defines the *spectral graph convolutional* operator to learn the nodes' representations as follows:

$$\mathbf{H} = \text{SGC}(\mathbf{X}; G, \mathbf{W}) = \text{ReLU}\left(\hat{\mathbf{A}} \mathbf{X} \mathbf{W}\right), \tag{1}$$

where $\mathbf{W} \in \mathbb{R}^{d_x \times d_h}$ is the variable involved in the operator.

Furthermore, GCN can involve a deep architecture by stacking multiple *spectral graph convolutional* layers on each other, which will be able to learn very complex high-level representations of the nodes. Here, let's assume the model depth to be $K$ (i.e., the number of hidden layers and output layer), and the corresponding node representation updating equations can be denoted as:

$$\begin{cases} \mathbf{H}^{(0)} & = \mathbf{X}, \\ \mathbf{H}^{(k)} & = \text{ReLU}\left(\hat{\mathbf{A}} \mathbf{H}^{(k-1)} \mathbf{W}^{(k-1)}\right), \forall k \in \{1, 2, \cdots, K-1\}, \\ \hat{\mathbf{Y}} & = \text{softmax}\left(\hat{\mathbf{A}} \mathbf{H}^{(K-1)} \mathbf{W}^{(K-1)}\right). \end{cases} \tag{2}$$

---

[1]https://paperswithcode.com/task/node-classification

## 3.2 Suspended Animation Problem with GCN

By investigating the *spectral graph convolutional* operator defined above, we observe that it actually involves two sequential steps:

$$\mathbf{H}^{(k)} = \text{ReLU}\left(\hat{\mathbf{A}}\mathbf{H}^{(k-1)}\mathbf{W}^{(k-1)}\right) \Leftrightarrow \begin{cases} \text{MC Layer:} & \mathbf{T}^{(k)} = \hat{\mathbf{A}}\mathbf{H}^{(k-1)} \\ \text{FC Layer:} & \mathbf{H}^{(k)} = \text{ReLU}\left(\mathbf{T}^{(k)}\mathbf{W}^{(k-1)}\right), \end{cases} \quad (3)$$

where the first term on the right-hand-side defines a 1-step Markov chain (MC or a random walk) based on the graph and the second term is a fully-connected (FC) layer parameterized by variable $\mathbf{W}^{(k-1)}$. Similar observations have been reported in Li et al. (2018b) as well, but it interprets the *spectral graph convolutional* operator in a different way as the *Laplacian smoothing* operator used in mesh smoothing in graphics instead.

Therefore, stacking multiple *spectral graph convolutional* layers on top of each other is equivalent to the stacking of multiple 1-step Markov chain layers and fully-connected layers in a crosswise way. Considering that the variables $\mathbf{W}^{(k-1)}$ for the vector dimension adjustment are shared among all the nodes, given two nodes with identical representations, the fully-connected layers (parameterized by $\mathbf{W}^{(k-1)}$) will still generate identical representations as well. In other words, the fully-connected layers with shared variables for all the nodes will not have significant impacts on the convergence of Markov chain layers actually. Therefore, in the following analysis, we will simplify the model structure by assuming the mapping defined by fully-connected layers to be the identity mapping. We will investigate the Markov chain layers closely by picking them out of the model to compose the Markov chain of multiple steps:

$$\begin{cases} \mathbf{T}^{(0)} & = \mathbf{X}, \\ \mathbf{T}^{(k)} & = \hat{\mathbf{A}}\mathbf{T}^{(k-1)}, \forall k \in \{1, 2, \cdots, K\}. \end{cases} \quad (4)$$

Meanwhile, the Markov chain layers may converge with $k$ layers iff $\mathbf{T}^{(k)} = \mathbf{T}^{(k-1)}$, i.e., the representations before and after the updating are identical (or very close), which is highly dependent on the input network structure, i.e., matrix $\hat{\mathbf{A}}$, actually.

**DEFINITION 1.** *(Irreducible and Aperiodic Network): Given an input network $G = (\mathcal{V}, \mathcal{E})$, $G$ is irreducible iff for any two nodes $v_i, v_j \in \mathcal{V}$, node $v_i$ is accessible to $v_j$. Meanwhile, $G$ is aperiodic iff $G$ is not bipartite.*

**LEMMA 1.** *Given an unweighted input graph $G$, which is irreducible, finite and aperiodic, if its corresponding matrix is asymmetric, starting from any initial distribution vector $\mathbf{x} \in \mathbb{R}^{n \times 1}$ ($\mathbf{x} \geq \mathbf{0}$ and $\|\mathbf{x}\|_1 = 1$), the Markov chain operating on the graph has one unique stationary distribution vector $\boldsymbol{\pi}^*$ such that $\lim_{t \to \infty} \hat{\mathbf{A}}^t \mathbf{x} = \boldsymbol{\pi}^*$, where $\boldsymbol{\pi}^*(i) = \frac{d(v_i)}{2|\mathcal{E}|}$. Meanwhile, if matrix $\hat{\mathbf{A}}$ is symmetric, the stationary distribution vector $\boldsymbol{\pi}^*$ will be a uniform distribution over the nodes, i.e., $\boldsymbol{\pi}^*(i) = \frac{1}{n}$.*

Based on the above Lemma 1, we can derive similar results for the multiple Markov chain layers in the GCN model based on the nodes' feature inputs, which will reduce the learned nodes' representations to the stationary representation matrix.

**THEOREM 1.** *Given a input network $G = (\mathcal{V}, \mathcal{E})$, which is unweighted, irreducible, finite and aperiodic, if there exist enough nested Markov chain layers in the GCN model, it will reduce the nodes' representations from the column-normalized feature matrix $\mathbf{X} \in \mathbb{R}^{n \times d_x}$ to the stationary representation $\boldsymbol{\Pi}^* = [\boldsymbol{\pi}^*, \boldsymbol{\pi}^*, \cdots, \boldsymbol{\pi}^*] \in \mathbb{R}^{n \times d_x}$. Furthermore, if $G$ is undirected, then the stationary representation will become $\boldsymbol{\Pi}^* = \frac{1}{n} \cdot \mathbf{1}^{n \times d_x}$.*

Theorem 1 illustrates the causes of the GNNs' suspended animation problem. Proofs of Lemma 1 and Theorem 1 are provided in the appendix attached to this paper at the end.

## 4 Suspended Animation Limit Analysis

Here, we will study the *suspended animation limit* of the GCN model based on its spectral convolutional operator analysis, especially the Markov chain layers.

### 4.1 Suspended Animation Limit based Input Network Structure

Formally, we define the *suspended animation limit* of GCN as follows:

**DEFINITION 2.** *(Suspended Animation Limit): The suspended animation limit of* GCN *on network* $G$ *is defined as the smallest model depth* $\tau$ *such that for any nodes' column-normalized featured matrix input* $\mathbf{X}$ *in the network* $G$ *the following inequality holds:*

$$\|GCN(\mathbf{X};\tau) - \mathbf{\Pi}^*\|_1 \leq \epsilon. \tag{5}$$

*For representation convenience, we can also denote the suspended animation limit of* GCN *defined on network* $G$ *as* $\zeta(G)$ *(or* $\zeta$ *for simplicity if there is no ambiguity problems).*

Based on the above definition, for GCN with identity FC mappings, there exists a tight bound of the *suspended animation limit* for the input network.

**THEOREM 2.** *Let* $1 \geq \lambda_1 \geq \lambda_2 \geq \cdots \geq \lambda_n$ *be the eigen-values of matrix* $\hat{\mathbf{A}}$ *defined based on network* $G$, *then the corresponding suspended animation limit of the* GCN *model on* $G$ *is bounded*

$$\zeta \leq \mathcal{O}\left(\frac{\log \min_i \frac{1}{\pi^*(i)}}{1 - \max\{\lambda_2, |\lambda_n|\}}\right). \tag{6}$$

*In the case that the network* $G$ *is a d-regular, then the suspended animation limit of the* GCN *model on* $G$ *can be simplified as*

$$\zeta \leq \mathcal{O}\left(\frac{\log n}{1 - \max\{\lambda_2, |\lambda_n|\}}\right). \tag{7}$$

The *suspended animation limit* bound derived in the above theorem generally indicates that the network structure $G$ determines the maximum allows depth of GCN. Among all the eigen-values of $\hat{\mathbf{A}}$ defined on network $G$, $\lambda_2$ measures how far $G$ is from being disconnected; and $\lambda_n$ measures how far $G$ is from being bipartite. In the case that $G$ is *reducible* (i.e., $\lambda_2 = 1$) or *bipartite* (i.e., $\lambda_n = -1$), we have $\zeta \to \infty$ and the model will not suffer from the *suspended animation problem*.

In the appendix of Kipf & Welling (2016), the authors introduce a naive-residual based variant of GCN, and the *sepctral graph convolutional* operator is changed as follows (the activation function is also changed to sigmoid function instead):

$$\mathbf{H}^{(k)} = \sigma\left(\hat{\mathbf{A}}\mathbf{H}^{(k-1)}\mathbf{W}^{(k-1)}\right) + \mathbf{H}^{(k-1)}, \tag{8}$$

For the identity fully-connected layer mapping, the above equation can be reduced to the following *lazy Markov chain* based layers

$$\mathbf{H}^{(k)} = 2 \cdot \left(\frac{1}{2}\hat{\mathbf{A}}\mathbf{H}^{(k-1)} + \frac{1}{2}\mathbf{H}^{(k-1)}\right). \tag{9}$$

Such a residual term will not help resolve the problem, and it will still suffer from the suspended animation problem with the following suspended animation limit bound:

**COROLLARY 1.** *Let* $1 \geq \lambda_1 \geq \lambda_2 \geq \cdots \geq \lambda_n$ *be the eigen-values of matrix* $\hat{\mathbf{A}}$ *defined based on network* $G$, *then the corresponding suspended animation limit of the* GCN *model (with lazy Markov chain based layers) on* $G$ *is bounded*

$$\zeta \leq \mathcal{O}\left(\frac{\log \min_i \frac{1}{\pi^*(i)}}{1 - \lambda_2}\right). \tag{10}$$

Proofs to Theorem 2 and Corollary 1 will be provided in the appendix as well.

### 4.2 OTHER PRACTICAL FACTORS IMPACTING THE SUSPENDED ANIMATION LIMIT

According to our preliminary experimental studies, GCN is quite a sensitive model. Besides the impact of input network explicit structures (e.g., network size $n$, directed vs undirected links, and network eigenvalues) as indicated in the bound equation in Theorem 2, many other factors can also influence the performance of GCN model a lot, which are summarized as follows:

- **Network Degree Distribution**: According to Theorem 1, if the input network $G$ is directed and unweighted, the Markov chain layers at convergence will project the features to $\mathbf{\Pi}^* =$

Table 1: A Summary of Graph Residual Terms and Physical Meanings.

| Name | Residual Term | Description |
|---|---|---|
| naive residual | $R\left(\mathbf{H}^{(k-1)}, \mathbf{X}; G\right) = \mathbf{H}^{(k-1)}$ | Node residual terms are assumed to be independent and determined by the current state only. |
| graph-naive residual | $R\left(\mathbf{H}^{(k-1)}, \mathbf{X}; G\right) = \hat{\mathbf{A}}\mathbf{H}^{(k-1)}$ | Node residual terms are correlated based on network structure, and can be determined by the current state. |
| raw residual | $R\left(\mathbf{H}^{(k-1)}, \mathbf{X}; G\right) = \mathbf{X}$ | Node residual terms are assumed to be independent and determined by the raw input features only. |
| graph-raw residual | $R\left(\mathbf{H}^{(k-1)}, \mathbf{X}; G\right) = \hat{\mathbf{A}}\mathbf{X}$ | Node residual terms are correlated based on network structure, and are determined by the raw input features. |

$[\boldsymbol{\pi}^*, \boldsymbol{\pi}^*, \cdots, \boldsymbol{\pi}^*]$, where $\boldsymbol{\pi}^*(i)$ is determined by the degree of node $v_i$ in the network. For any two nodes $v_i, v_j \in \mathcal{V}$, the differences of their learned representation can be denoted as

$$d(v_i, v_j) = \|\boldsymbol{\Pi}(i,:) - \boldsymbol{\Pi}(j,:)\|_1 = \sum_{k=1}^{d_x} |\boldsymbol{\Pi}(i,k) - \boldsymbol{\Pi}(i,k)| = d_x \frac{|d(v_i) - d(v_j)|}{2|\mathcal{E}|}. \quad (11)$$

According to Newman (2003), the node degree distributions in most complex networks follow the power-law Faloutsos et al. (1999), i.e., majority of the nodes are of very small degrees. Therefore, for massive nodes in the input network with the same (or close) degrees, the differences between their learned representations will become not distinguishable.

- **Raw Feature Coding**: Besides the input network, the raw feature coding can also affect the learning performance greatly. Here, we can take the GCN with one single Markov chain layer and one identity mapping layer. For any two nodes $v_i, v_j \in \mathcal{V}$ with raw feature vectors $\mathbf{X}(i,:)$ and $\mathbf{X}(j,:)$, we can denote the differences between their learned representations as follows:

$$d(v_i, v_j) = \|\mathbf{T}(i,:) - \mathbf{T}(j,:)\|_1 = \left\|\left(\hat{\mathbf{A}}(i,:) - \hat{\mathbf{A}}(j,:)\right) \mathbf{X}\right\|_1. \quad (12)$$

Different from Equation (11), analysis in the above equation is not based on the stationary representation of the nodes, and it is based on the MC-layer representation as indicated in Equation (3) and Equation (4). For the one-hot feature coding used in the source code of GCN Kipf & Welling (2016) and other GNNs, matrix $\mathbf{X}$ can be also very sparse as well. Meanwhile, vector $\hat{\mathbf{A}}(i,:) - \hat{\mathbf{A}}(j,:)$ is also a sparse vector, which renders the right-hand-side term to be a very small value.

- **Training Set Size**: Actually, the nodes have identical representation and the same labels will not degrade the learning performance of the model. However, if such node instances actually belong to different classes, it will become a great challenge for both the training and test stages of the GCN model.

- **Gradient Vanishing/Exploding**: Similar to the existing deep models, deep GNNs will also suffer from the gradient vanishing/exploding problems Pascanu et al. (2012), which will also greatly affect the learning performance of the models.

Although these factors mentioned above are not involved in the *suspended animation limit* bound representation, but they do have great impacts on the GCN model in practical applications. In the following section, we will introduce *graph residual network* GRESNET, which can be useful for resolving such a problem for GCN.

## 5 GRAPH RESIDUAL NETWORK

Different from the residual learning in other areas, e.g., computer vision He et al. (2015), where the objective data instances are independent with each other, in the inter-connected network learning setting, the residual learning of the nodes in the network are extensively connected instead. It renders the existing residual learning strategy less effective for improving the performance of GCN.

### 5.1 GRAPH RESIDUAL LEARNING

Residual learning initially introduced in He et al. (2015) divides the objective mapping into two parts: the inputs and the residual function. For instance, let $H(\mathbf{x})$ be the objective mapping which projects input $\mathbf{x}$ to the desired domain. The ResNet introduced in He et al. (2015) divides $H(\mathbf{x})$ as $F(\mathbf{x}) + R(\mathbf{x})$ (where $R(\mathbf{x}) = \mathbf{x}$ is used in He et al. (2015)). This reformulation is motivated by the counterintuitive phenomena about the degradation problem observed on the deep CNN. Different

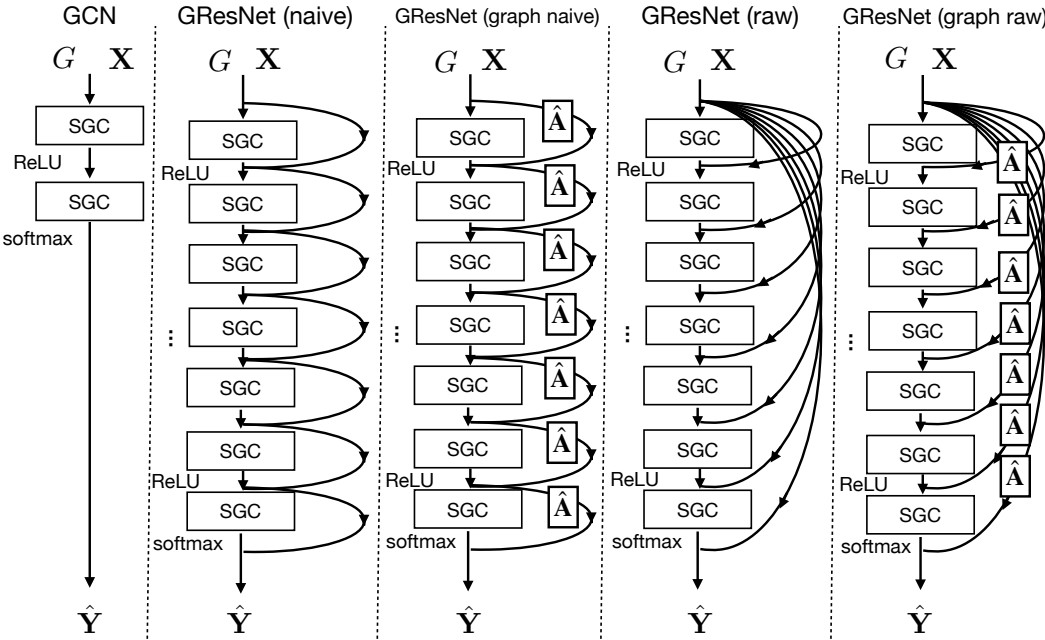

Figure 2: A comparison of vanilla GCN and GRESNET (GCN) with different graph residual terms. The vanilla GCN has two layers, and the GRESNET models have a deep architecture which involves seven layers of SGC operators. The intermediate ReLU activation functions used between sequential layers in GRESNET are omitted for simplicity. Notation $\boxed{\hat{\mathbf{A}}}$ denotes the normalized adjacency matrix of input network $G$, which indicates the correlated graph residual terms.

from the learning settings of CNN, where the data instances are assumed to be independent, the nodes inside the input network studied in GCN are closely correlated. Viewed in such a perspective, new residual learning mechanism should be introduced for GCN specifically.

**Here, we need to add a remark**: For the presentation simplicity in this paper, given the objective $H(\mathbf{x}) = F(\mathbf{x}) + R(\mathbf{x})$, we will misuse the terminologies here: we will name $F(\mathbf{x})$ as the approximated mapping of $H(\mathbf{x})$, and call $R(\mathbf{x})$ as the *graph residual term*. Formally, by incorporating the residual learning mechanism into the GCN model, the node representation updating equations (i.e., Equation 2) can be rewritten as follows:

$$\begin{cases} \mathbf{H}^{(0)} & = \mathbf{X} \\ \mathbf{H}^{(k)} & = \text{ReLU}\left(\hat{\mathbf{A}}\mathbf{H}^{(k-1)}\mathbf{W}^{(k-1)} + \text{R}\left(\mathbf{H}^{(k-1)}, \mathbf{X}; G\right)\right), \forall k \in \{1, 2, \cdots, K-1\}, \\ \hat{\mathbf{Y}} & = \text{softmax}\left(\hat{\mathbf{A}}\mathbf{H}^{(K-1)}\mathbf{W}^{(K)} + \text{R}\left(\mathbf{H}^{(K-1)}, \mathbf{X}; G\right)\right). \end{cases} \quad (13)$$

The *graph residual term* $\text{R}\left(\mathbf{H}^{(k-1)}, \mathbf{X}; G\right), \forall k \in \{1, 2, \cdots, K\}$ can be defined in different ways. We have also examined to put $\text{R}\left(\mathbf{H}^{(k-1)}, \mathbf{X}; G\right)$ outside of the ReLU$(\cdot)$ function for the hidden layers (i.e., $\mathbf{H}^{(k)} = \text{ReLU}\left(\hat{\mathbf{A}}\mathbf{H}^{(k-1)}\mathbf{W}^{(k-1)}\right) + \text{R}\left(\mathbf{H}^{(k-1)}, \mathbf{X}; G\right)$), whose performance is not as good as what we show above. In the appendix of Kipf & Welling (2016), by following the ResNet (CNN) He et al. (2015), the *graph residual term* $\text{R}\left(\mathbf{H}^{(k-1)}, \mathbf{X}; G\right)$ is simply defined as $\mathbf{H}^{(k-1)}$, which is named as the *naive residual term* in this paper (Here, term "naive" has no disparaging meanings). However, according to the studies, such a simple and independent residual term for the nodes fail to capture information in the inter-connected graph learning settings.

## 5.2 GRESNET ARCHITECTURE

In this paper, we introduce several other different representations of the *graph residual term*, which are summarized in Table 1. If feature dimension adjustment is needed, e.g., for raw residual term, an extra variable matrix $\mathbf{W}^{adj}$ can be added to redefine the terms (which are not shown in this paper). For the graph residual term representations in Table 1, *naive residual* and *raw residual* are based on the assumption that node residuals are independent and determined by either the current state or the raw features. Meanwhile, the *graph naive residual* and *graph raw residual* assume the residual terms of different nodes are correlated instead, which can be computed with the current state or raw features. We also illustrate the architectures of vanilla 2-layer GCN and the 7-layer GRESNETs (taking GCN as the base model) with different *graph residual terms* in Figure 2. We

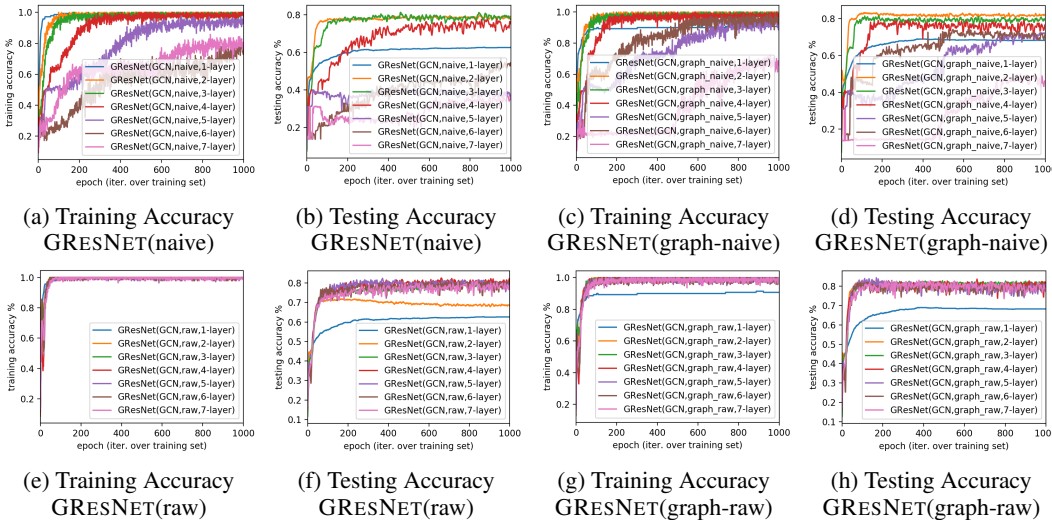

(a) Training Accuracy GRESNET(naive)  (b) Testing Accuracy GRESNET(naive)  (c) Training Accuracy GRESNET(graph-naive)  (d) Testing Accuracy GRESNET(graph-naive)

(e) Training Accuracy GRESNET(raw)  (f) Testing Accuracy GRESNET(raw)  (g) Training Accuracy GRESNET(graph-raw)  (h) Testing Accuracy GRESNET(graph-raw)

Figure 3: The learning performance of GRESNET with GCN as the base model and different graph residual terms on the Cora dataset: (a)-(b) GRESNET(GCN, naive); (c)-(d) GRESNET(GCN, graph-naive); (e)-(f) GRESNET(GCN, raw); (g)-(h) GRESNET(GCN, graph-raw). For plot (b), the curves corresponding to the 5-layer, 6-layer and 7-layer models are hidden by the legend. For plot (c) and (d), the curve of 7-layer model is hidden by the legend.

need to add a remark here that similar solutions to the raw residual terms have been studied in the latest research papers Wang et al. (2018); Wang & Gupta (2018) as well. However, in this paper, we aim to generalize such techniques as a unified framework, which can accept different types of residual terms not just the raw residual term.

**Vanilla GCN**: For the vanilla GCN network used in Kipf & Welling (2016), i.e., the left plot of Figure 2, given the inputs $G$ and $\mathbf{X}$, it employs two SGC layers to project the input to the objective labels. For the intermediate representations, the hidden layer length is 16, and ReLU is used as the activation function, whereas softmax is used for output label normalization.

**GRESNET Network**: For the GRESNET network, i.e., the right four models in Figure 2, they accept the identical inputs as vanilla GCN but will create *graph residual terms* to be added to the intermediate representations. Depending on the specific residual term representations adopted, the corresponding high-way connections can be different. For the hidden layers involved in the models, their length is also 16, and ReLU is used as the activation function for the intermediate layers.

By comparing the output $\hat{\mathbf{Y}}$ learned by the models against the ground-truth $\mathbf{Y}$ of the training instances, all the variables involved in the model, i.e., $\boldsymbol{\Theta}$, can be effectively learned with the back-propagation algorithm to minimize the loss functions $\ell(\mathbf{Y}, \hat{\mathbf{Y}}; \boldsymbol{\Theta})$ (or $\ell(\boldsymbol{\Theta})$ for simplicity). In the following part, we will demonstrate that for the GRESNET model added with graph residual terms. It can effectively avoid dramatic changes to the nodes' representations between sequential layers.

## 5.3 GRAPH RESIDUAL LEARNING EFFECTIVENESS ANALYSIS

In this part, we will illustrate why the inclusion of the *graph residual learning* can be effective for learning deep graph neural networks. Here, we assume the ultimate model that we want to learn as $H : \mathcal{X} \to \mathcal{Y}$, where $\mathcal{X}$ and $\mathcal{Y}$ denote the feature and label spaces, respectively. For analysis simplicity, we have some assumptions about the function $H$ Zaeemzadeh et al. (2018).

**ASSUMPTION 1.** *Function $H$ is differentiable, invertible and satisfies the following conditions:*

- $\forall \mathbf{x}, \mathbf{y}, \mathbf{z} \in \mathcal{X}$ *with bounded norm,* $\exists \alpha > 0$, $\|(H'(\mathbf{x}) - H'(\mathbf{y}))\mathbf{z}\| \le \alpha \cdot \|\mathbf{x} - \mathbf{y}\| \cdot \|\mathbf{z}\|$,
- $\forall \mathbf{x}, \mathbf{y} \in \mathcal{X}$ *with bounded norm,* $\exists \beta > 0$, $\|H^{-1}(\mathbf{x}) - H^{-1}(\mathbf{y})\| \le \beta \cdot \|\mathbf{x} - \mathbf{y}\|$,
- $\exists \mathbf{x} \in \mathcal{X}$ *with bounded norm such that* $Det(H'(\mathbf{x})) > 0$.

*In the above conditions, terms $\alpha$ and $\beta$ are constants.*

To model the function $H$, the GRESNET actually defines a sequence of $K$ sequential mappings with these $K$ layers:

$$\mathbf{x}^{(k)} = F^{(k-1)}(\mathbf{x}^{(k-1)}) + R^{(k-1)}(\mathbf{x}^{(k-1)}), \tag{14}$$

where $\mathbf{x}^{(k-1)}$ and $\mathbf{x}^{(k)}$ denote the intermediate node representations serving as input and output of the $k_{th}$ layer. $F^{(k-1)}(\cdot)$ and $R^{(k-1)}(\cdot)$ denote the function approximation and residual term

Table 2: Best performance (accuracy) and model depth summarization of GRESNET with different residual terms on the benchmark datasets (we take GCN, GAT and LOOPYNET as the base models).

| Methods | | Datasets (Accuracy & Model Depth) | | | | | |
|---|---|---|---|---|---|---|---|
| Base Models | Residuals | **Cora** | | **Citeseer** | | **Pubmed** | |
| vanilla GCN (Kipf & Welling (2016)) | | 0.815 | 2-layer | 0.703 | 2-layer | 0.790 | 2-layer |
| GCN | naive | 0.814 | 3-layer | 0.710 | 3-layer | 0.814 | 3-layer |
| | graph-naive | 0.833 | 2-layer | 0.715 | 3-layer | 0.811 | 2-layer |
| | raw | 0.826 | 4-layer | **0.727** | **4-layer** | 0.810 | 3-layer |
| | graph-raw | **0.843** | **5-layer** | 0.722 | 4-layer | **0.817** | **7-layer** |
| vanilla GAT (Veličković et al. (2018)) | | 0.830 | 2-layer | 0.725 | 2-layer | 0.790 | 2-layer |
| GAT | naive | 0.844 | 5-layer | **0.735** | **5-layer** | 0.809 | 3-layer |
| | graph-naive | **0.855** | **3-layer** | 0.732 | 4-layer | 0.815 | 5-layer |
| | raw | 0.842 | 3-layer | 0.733 | 3-layer | 0.814 | 4-layer |
| | graph-raw | 0.847 | 3-layer | 0.729 | 5-layer | **0.822** | **4-layer** |
| vanilla LOOPYNET (Zhang (2018)) | | 0.826 | 2-layer | 0.716 | 2-layer | 0.812 | 2-layer |
| LOOPYNET | naive | 0.833 | 2-layer | 0.728 | 3-layer | **0.830** | **4-layer** |
| | graph-naive | 0.832 | 2-layer | 0.728 | 3-layer | 0.819 | 2-layer |
| | raw | 0.836 | 2-layer | 0.730 | 5-layer | 0.828 | 4-layer |
| | graph-raw | **0.839** | **4-layer** | **0.737** | **5-layer** | 0.814 | 4-layer |

learned by the $k - 1_{th}$ layer of the model. When training these $K$ sequential mappings, we have the following theorem hold for the representation gradients in the learning process.

**THEOREM 3.** *Let $H$ denote the objective function that we want to model, which satisfies Assumption 1, in learning the $K$-layer GRESNET model, we have the following inequality hold:*

$$(1 - \delta) \left\| \frac{\partial \ell(\mathbf{\Theta})}{\partial \mathbf{x}^{(k)}} \right\|_2 \leq \left\| \frac{\partial \ell(\mathbf{\Theta})}{\partial \mathbf{x}^{(k-1)}} \right\|_2 \leq (1 + \delta) \left\| \frac{\partial \ell(\mathbf{\Theta})}{\partial \mathbf{x}^{(k)}} \right\|_2, \tag{15}$$

*where $\delta \leq c \cdot \frac{log(2K)}{K}$ and $c = c_1 \cdot \max\{\alpha \cdot \beta \cdot (1 + \beta), \beta \cdot (2 + \alpha) + \alpha\}$ for some $c_1 > 0$.*

Proof of Theorem 3 will be provided in the appendix. The above theorem indicates that in the learning process, the norm of loss function against the intermediate representations doesn't change significantly between sequential layers. In other words, GRESNET can maintain effective representations for the inputs and overcome the *suspended animation problem*. In addition, we observe that the bound of the gap term $\delta$, i.e., $c \cdot \frac{log(2K)}{K}$, decreases as $K$ increases (when $K \geq 2$). Therefore, for deeper GRESNET, the model will lead to much tighter gradient norm changes, which is a desired property. In the following section, we will provide the experimental results of GRESNET compared against their vanilla models on several graph benchmark datasets.

## 6 EXPERIMENTS

To demonstrate the effectiveness of GRESNET in improving the learning performance for graph neural networks with deep architectures, extensive experiments will be done on several graph benchmark datasets. Similar to the previous works on node classification Kipf & Welling (2016); Veličković et al. (2018), the graph benchmark datasets used in the experiments include Cora, Citeseer and Pubmed from Sen et al. (2008). For fair comparison, we follow exactly the same experimental settings as Kipf & Welling (2016) on these datasets.

In this paper, we aim at studying the suspended animation problem with the existing graph neural networks, e.g., GCN Kipf & Welling (2016), GAT Veličković et al. (2018) and LOOPYNET Zhang (2018), where LOOPYNET is not based on the spectral graph convolutional operator. We also aim to investigate the effectiveness of these proposed graph residual terms in improving their learning performance, especially for the models with deep architectures. In addition, to make the experiments self-contained, we also provide the latest performance of the other baseline methods on the same datasets in this paper, which include state-of-the-art graph neural networks, e.g., APPNP Klicpera et al. (2019), GOCN Jiang et al. (2019) and GraphNAS Gao et al. (2019), existing graph embedding models, like DeepWalk Perozzi et al. (2014), Planetoid Yang et al. (2016) and MoNet Monti et al. (2016) and representation learning approaches, like ManiReg Belkin et al. (2006), SemiEmb Weston et al. (2008), LP Zhu et al. (2003) and ICA Lu & Getoor (2003).

Table 3: Learning result accuracy of node classification methods. In the table, '-' denotes the results of the methods on these datasets are not reported in the existing works. Performance of GCN, GAT and LOOPYNET shown in Table 2 are not provided here to avoid reporting duplicated results.

| Methods | Datasets (Accuracy) | | |
|---|---|---|---|
| | **Cora** | **Citeseer** | **Pubmed** |
| LP (Zhu et al. (2003)) | 0.680 | 0.453 | 0.630 |
| ICA (Lu & Getoor (2003)) | 0.751 | 0.691 | 0.739 |
| ManiReg (Belkin et al. (2006)) | 0.595 | 0.601 | 0.707 |
| SemiEmb (Weston et al. (2008)) | 0.590 | 0.596 | 0.711 |
| DeepWalk (Perozzi et al. (2014)) | 0.672 | 0.432 | 0.653 |
| Planetoid (Yang et al. (2016)) | 0.757 | 0.647 | 0.772 |
| MoNet (Monti et al. (2016)) | 0.817 | - | 0.788 |
| APPNP (Klicpera et al. (2019)) | **0.851** | **0.757** | 0.797 |
| GOCN (Jiang et al. (2019)) | **0.848** | 0.718 | 0.797 |
| GraphNAS (Gao et al. (2019)) | - | 0.731 | 0.769 |
| GRESNET(GCN) | 0.843 | 0.727 | **0.817** |
| GRESNET(GAT) | **0.855** | 0.735 | **0.822** |
| GRESNET(LOOPYNET) | 0.839 | **0.737** | **0.830** |

**Reproducibility**: Both the datasets and source code used in this paper can be accessed via link[2]. Detailed information about the server used to run the model can be found at the footnote[3].

## 6.1 EFFECTIVENESS OF THE GRAPH RESIDUAL TERMS

In addition to Figure 1 for GCN (bias disabled) on the Cora dataset, as shown in Figures 9-11 in the appendix, for the GCN (bias enabled) and GAT with deep architectures, we have observed similar suspended animation problems. Meanwhile, the performance of LOOPYNET is different. Since LOOPYNET is not based on the spectral graph convolution operator, which accepts nodes' raw features in all the layer (it is quite similar to the *raw* residual term introduced in this paper). As the model depth increase, performance of LOOPYNET remains very close but converge much more slowly. By taking GCN as the base model, we also show the performance of GRESNET with different residual terms in Figure 3. By comparing these plots with Figure 1, both *naive* and *graph-naive* residual terms help stabilize the performance of deep GRESNET(GCN)s. Meanwhile, for the *raw* and *graph-raw* residual terms, their contributions are exceptional. With these two residual terms, deep GRESNET(GCN)s can achieve even better performance than the shallow vanilla GCNs.

Besides the results on the Cora dataset, in Table 2, we illustrate the best observed performance by GRESNET with different residual terms based on GCN, GAT and LOOPYNET base models respectively on all the datasets. Both the best accuracy score and the achieved model depth are provided. According to the results, for the vanilla models, GCN, GAT and LOOPYNET can all obtain the best performance with shallow architectures. For instance on Cora, GCN(2-layer) obtains 0.815; GAT(2-layer) gets 0.830; and LOOPYNET(2-layer) achieves 0.826, respectively. Added with the residual terms, the performance of all these models will get improved. In addition, deep GRESNET(GCN), GRESNET(GAT) and GRESNET(LOOPYNET) will be able to achieve much better results than the shallow vanilla models, especially the ones with the *graph-raw residual* terms. The best scores and the model depth for each base model on these datasets are also highlighted. The time costs of learning the GRESNET model is almost identical to the required time costs of learning the vanilla models with the same depth, which are not reported in this paper.

## 6.2 A COMPLETE COMPARISON WITH EXISTING NODE CLASSIFICATION METHODS

Besides the comparison with GCN, GAT and LOOPYNET shown in Table 2, to make the experimental studies more complete, we also compare GRESNET(GCN), GRESNET(GAT) and GRESNET(LOOPYNET) with both the classic and the state-of-the-art models, whose results are provided in Table 3. In the table, we didn't indicate the depth of the GRESNET models and results of GCN, GAT and LOOPYNET (shown in Table 2 already) are not included. According to the results, compared against these baseline methods, GRESNETs can also outperform them with great advantages. Without the complex model architecture extension or optimization techniques used by the latest methods APPNP Klicpera et al. (2019), GOCN Jiang et al. (2019) and GraphNAS Gao et al. (2019),

---

[2]https://github.com/anonymous-sourcecode/GResNet

[3]GPU Server: ASUS X99-E WS motherboard, Intel Core i7 CPU 6850K@3.6GHz (6 cores), 3 Nvidia GeForce GTX 1080 Ti GPU (11 GB buffer each), 128 GB DDR4 memory and 128 GB SSD swap. For the deep models which cannot fit in the GPU memory, we run them with CPU instead.

adding the simple graph residual terms into the base models along can already improve the learning performance greatly.

## 7 CONCLUSION

In this paper, we focus on studying the suspended animation problem with the existing graph neural network models, especially the spectral graph convolutional operator. We provide a theoretic analysis about the causes of the suspended animation problem and derive the bound for the maximum allowed graph neural network depth, i.e., the suspended animation limit. To resolve such a problem, we introduce a novel framework GRESNET, which works well for learning deep representations from the graph data. Assisted with these new graph residual terms, we demonstrate that GRESNET can effectively resolve the suspended animation problem with both theoretic analysis and empirical experiments on several benchmark node classification datasets.

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

## 8 REBUTTAL NEW MATERIALS

### 8.1 EXPERIMENTAL RESULTS ON OTHER LARGER DATASETS

We added the experimental results on the PPI and Reddit datasets in the following table. Detailed information about datasets is as follows:

- **PPI**: **Node #**: 14,755; **Edge #**: 225,270; **Feature #**: 50; **Class #**: 121
- **Reddit**: **Node #**: 232, 965; **Edge #**: 11,606,919; **Feature #**: 602; **Class #**: 41

We need to mention that GAT used in GResNet is very slow on large datasets (it is not our problem but the problem with the GAT base model), we cannot get the results of GResNet(GAT) out on Reddit during the rebuttal period. Since PPI and Reddit are not common used in other graph neural network papers, so many of the entries in Table 4 are not provided. We will add these results in the camera-ready version of this paper later.

Table 4: Learning result on larger datasets.

| Methods | Datasets (Accuracy) | | | Dataset (micro-averaged F1) | |
|---|---|---|---|---|---|
| | **Cora** | **Citeseer** | **Pubmed** | **PPI** | **Reddit** |
| LP (Zhu et al. (2003)) | 0.680 | 0.453 | 0.630 | - | - |
| ICA (Lu & Getoor (2003)) | 0.751 | 0.691 | 0.739 | - | - |
| ManiReg (Belkin et al. (2006)) | 0.595 | 0.601 | 0.707 | - | - |
| SemiEmb (Weston et al. (2008)) | 0.590 | 0.596 | 0.711 | - | - |
| DeepWalk (Perozzi et al. (2014)) | 0.672 | 0.432 | 0.653 | - | 0.324 |
| Planetoid (Yang et al. (2016)) | 0.757 | 0.647 | 0.772 | - | - |
| MoNet (Monti et al. (2016)) | 0.817 | - | 0.788 | - | - |
| APPNP (Klicpera et al. (2019)) | **0.851** | **0.757** | 0.797 | - | - |
| GOCN (Jiang et al. (2019)) | **0.848** | 0.718 | 0.797 | - | - |
| Graph-S (Gao et al. (2019)) | - | 0.731 | 0.769 | - | - |
| GCN (Kipf & Welling (2016)) | 0.815 | 0.703 | 0.790 | 0.500 | 0.930 |
| GAT (Veličković et al. (2018)) | 0.830 | 0.725 | 0.790 | **0.973** | - |
| LOOPYNET (Zhang (2018)) | 0.826 | 0.716 | 0.812 | - | - |
| GraphSAGE (Hamilton et al. (2017)) | - | - | - | 0.612 | **0.954** |
| FastGCN (Chen et al. (2018)) | 0.850 | - | 0.880 | 0.513 | 0.937 |
| GRESNET(GCN) | 0.843 | 0.727 | **0.817** | 0.832 | **0.950** |
| GRESNET(GAT) | **0.855** | 0.735 | 0.822 | **0.987** | - |
| GRESNET(LOOPYNET) | 0.839 | **0.737** | **0.830** | **0.980** | 0.955 |

## 8.2 EXPERIMENTAL RESULTS ON DEEPER MODELS

### 8.2.1 DEEPER MODEL PERFORMANCE SUMMARY

As suggested by the reviewers, we also add the studies of GCN, GRESNET(GCN, naive), GRES-NET(GCN, graph-naive), GRESNET(GCN, raw) and GRESNET(GCN, graph-raw) with deeper architectures (more than 7 layers) on the Cora dataset in the following table and plots

As illustrated in Table 7, we show the experimental results on Cora obtained by the models with 2, 10, 20, 30, 40, 50 layers, respectively. According to the numbers, GCN cannot really work well for all the layers other than 2, and the performance of the naive and graph-naive residual terms is not good neither. However, the performance of the raw and graph-raw residual terms is exceptionally great. For instance, for the model with 40 layers, GRESNET(GCN, raw) and GRESNET(GCN, graph-raw) can still achieve 0.790 and 0.838 as the accuracy scores.

Table 5: Learning result by models with deeper architectures.

| Methods | Model Depth (Accuracy) | | | | | |
|---|---|---|---|---|---|---|
| | 2 | 10 | 20 | 30 | 40 | 50 |
| GCN | **0.815** | 0.174 | 0.072 | 0.072 | 0.072 | 0.072 |
| GRESNET(GCN, naive) | 0.804 | 0.416 | 0.126 | 0.057 | 0.057 | 0.057 |
| GRESNET(GCN, graph-naive) | **0.833** | 0.169 | 0.057 | 0.057 | 0.057 | 0.057 |
| GRESNET(GCN, raw) | **0.815** | 0.814 | 0.802 | 0.812 | 0.798 | 0.800 |
| GRESNET(GCN, graph-raw) | 0.813 | **0.823** | **0.823** | **0.818** | **0.838** | **0.799** |

### 8.2.2 DEEPER MODEL LEARNING LOSS ON TRAINING AND TESTING SETS PER EPOCH

In addition to Table 7, we also illustrate the performance of the models in the learning process in Figures 4-8, where both the training accuracy and testing accuracy on each epoch are provided. According to the results, GCN and GRESNET(GCN, graph-naive) will fail to work when the model architecture goes deeper. GRESNET(GCN, naive) can slightly revive the models for 10 layers, but will also fail to work for deeper architectures, e.g., 20, 30, 40 and 50 layers. However, GRES-NET(GCN, raw) and GRESNET(GCN, graph-raw) work very great for all these different depth in the plot, which also illustrate the effectiveness of the raw and graph-raw residual terms proposed in this paper.

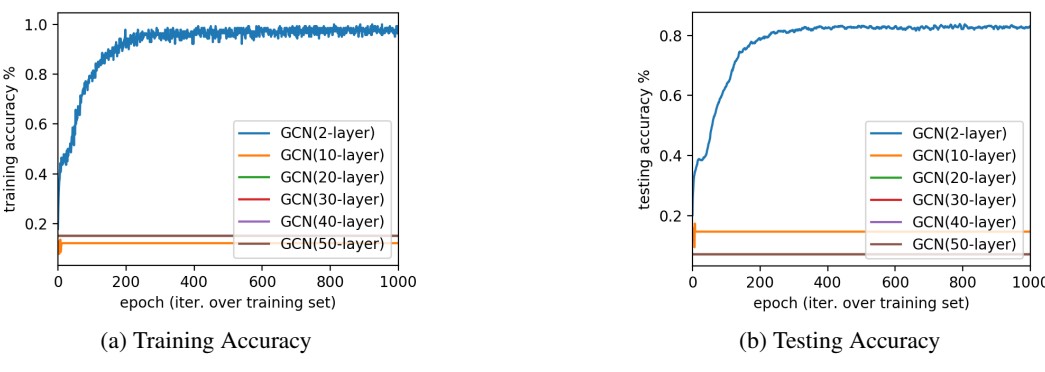

(a) Training Accuracy             (b) Testing Accuracy

Figure 4: Deeper GCN on Cora.

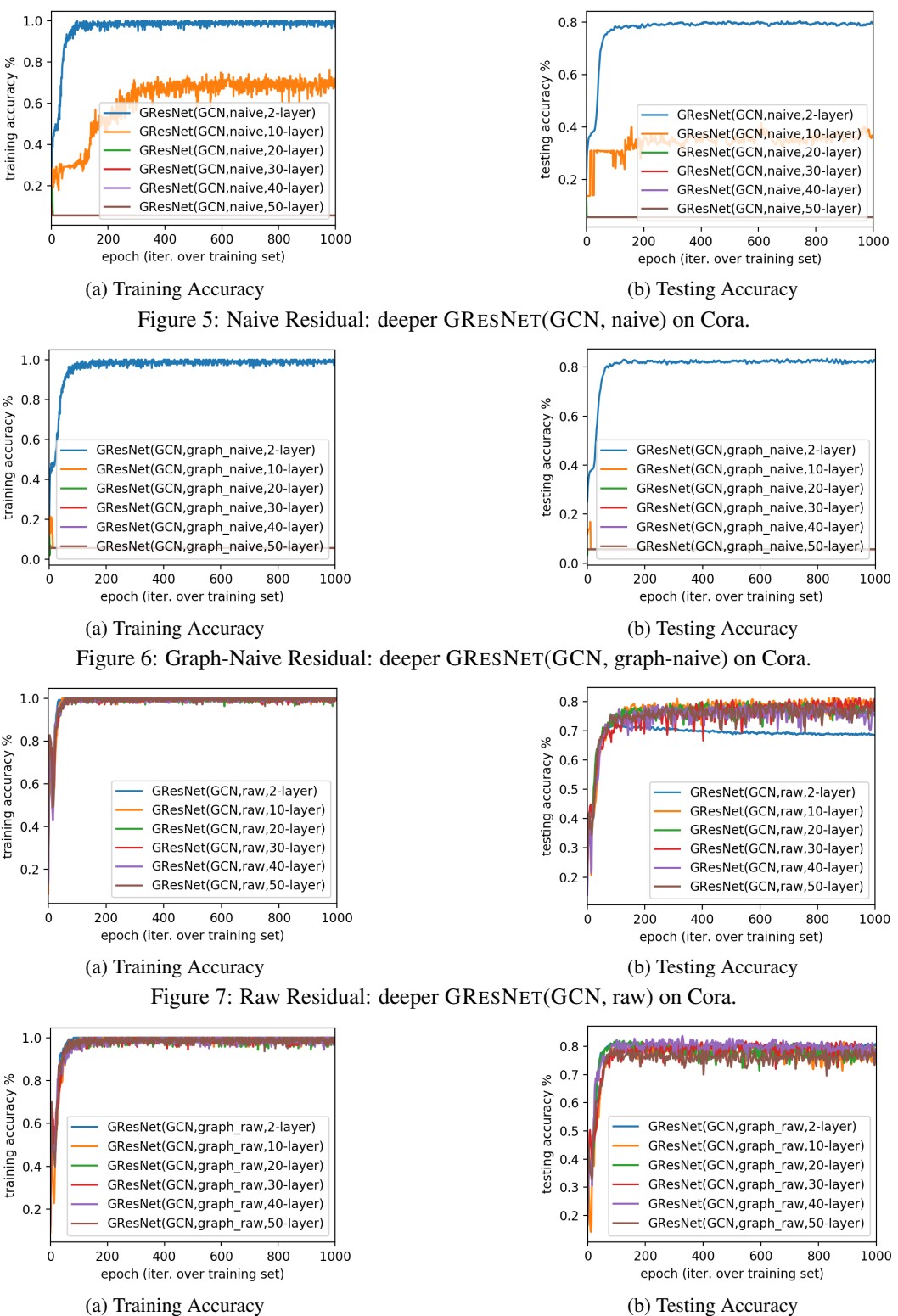

(a) Training Accuracy

(b) Testing Accuracy

Figure 5: Naive Residual: deeper GRESNET(GCN, naive) on Cora.

(a) Training Accuracy

(b) Testing Accuracy

Figure 6: Graph-Naive Residual: deeper GRESNET(GCN, graph-naive) on Cora.

(a) Training Accuracy

(b) Testing Accuracy

Figure 7: Raw Residual: deeper GRESNET(GCN, raw) on Cora.

(a) Training Accuracy

(b) Testing Accuracy

Figure 8: Graph-Raw Residual: deeper GRESNET(GCN, graph-raw) on Cora.

We have also studied even deeper model architectures on GRESNET(GCN, raw) and GRES-NET(GCN, graph-raw) to 100 layers, which can still perform well.

# 9 STUDIES ON OTHER IMPACTING FACTORS WITH RESIDUAL TERMS

By changes to the original raw data, we will be able to get different samples of the data with different properties, e.g., raw features, and the training set size. We will illustrate the learning results based on these different settings on the Cora dataset as follows for the reviewers' information.

## 9.1 TRAINING SET SIZE

Table 6: Learning result by models with deeper architectures.

| Methods | Training Set Size (Regular Size: 140) | | | |
|---|---|---|---|---|
| | 140 | 500 | 1000 | 1500 |
| GCN(2-layer) | 0.815 | 0.863 | 0.922 | 0.933 |
| GCN(5-layer) | 0.315 | 0.331 | 0.403 | 0.324 |
| GRESNET(GCN, naive, 2-layer) | 0.797 | 0.842 | 0.936 | 0.956 |
| GRESNET(GCN, naive, 5-layer) | 0.596 | 0.748 | 0.813 | 0.958 |
| GRESNET(GCN, graph-naive, 2-layer) | 0.833 | 0.862 | 0.927 | 0.956 |
| GRESNET(GCN, graph-naive, 5-layer) | 0.749 | 0.797 | 0.904 | 0.946 |
| GRESNET(GCN, raw, 2-layer) | 0.809 | 0.799 | 0.918 | 0.979 |
| GRESNET(GCN, raw, 5-layer) | 0.816 | 0.861 | 0.932 | 0.976 |
| GRESNET(GCN, graph-raw, 2-layer) | 0.809 | 0.853 | 0.929 | 0.966 |
| GRESNET(GCN, graph-raw, 5-layer) | 0.843 | 0.859 | 0.917 | 0.963 |

## 9.2 RAW FEATURE ENCODING

Table 7: Learning result by models with deeper architectures.

| Methods | Feature Coding (Regular Coding: bag-of-word) | |
|---|---|---|
| | bag-of-word | one-hot |
| GCN(2-layer) | 0.815 | 0.795 |
| GCN(5-layer) | 0.315 | 0.199 |
| GRESNET(GCN, naive, 2-layer) | 0.797 | 0.700 |
| GRESNET(GCN, naive, 5-layer) | 0.596 | 0.592 |
| GRESNET(GCN, graph-naive, 2-layer) | 0.833 | 0.797 |
| GRESNET(GCN, graph-naive, 5-layer) | 0.749 | 0.748 |
| GRESNET(GCN, raw, 2-layer) | 0.809 | 0.714 |
| GRESNET(GCN, raw, 5-layer) | 0.816 | 0.721 |
| GRESNET(GCN, graph-raw, 2-layer) | 0.809 | 0.746 |
| GRESNET(GCN, graph-raw, 5-layer) | 0.843 | 0.798 |

# 10 APPENDIX

## 10.1 EXTRA EXPERIMENTAL RESULTS

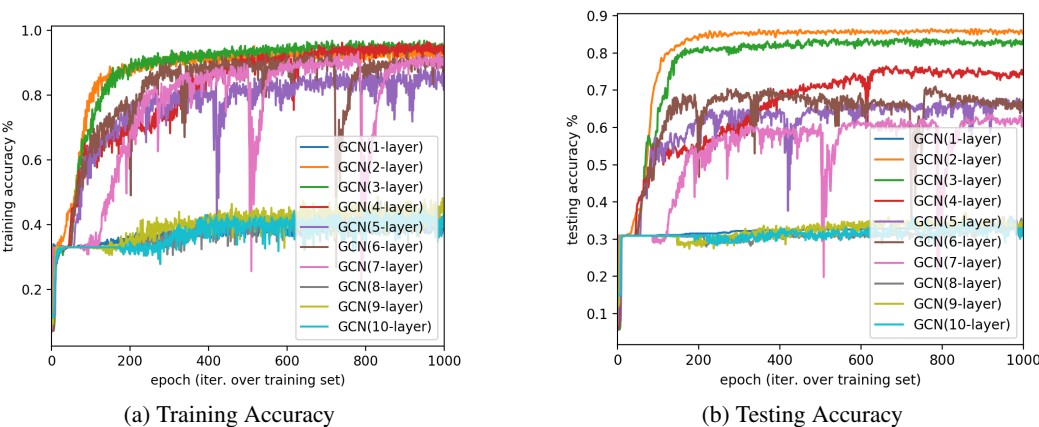

(a) Training Accuracy

(b) Testing Accuracy

Figure 9: The learning performance of GCN (bias enabled) on the Cora dataset.

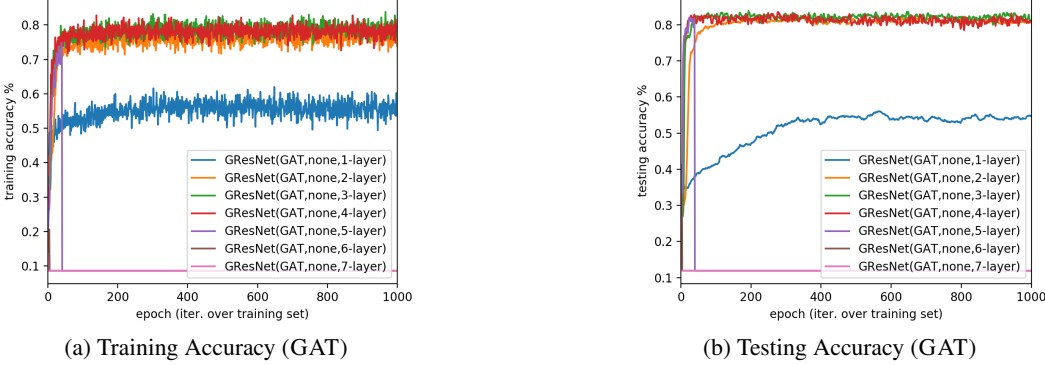

(a) Training Accuracy (GAT)

(b) Testing Accuracy (GAT)

Figure 10: The learning performance of GAT on the Cora dataset

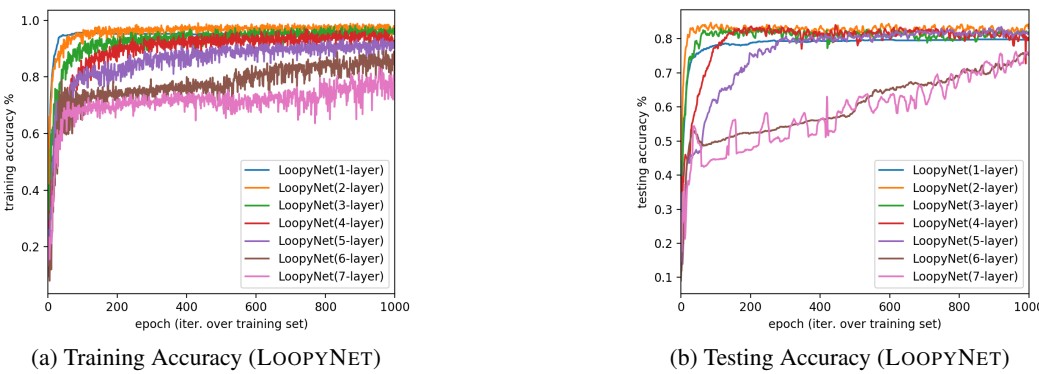

(a) Training Accuracy (LOOPYNET)

(b) Testing Accuracy (LOOPYNET)

Figure 11: The learning performance of LOOPYNET on the Cora dataset

## 10.2 PROOFS OF THEOREM AND LEMMA

### 10.2.1 PROOF OF LEMMA 1

**LEMMA 1.** *Given an irreducible, finite and aperiodic graph $G$, starting from any initial distribution vector $\mathbf{x} \in \mathbb{R}^{n \times 1}$ ($\mathbf{x} \geq \mathbf{0}$ and $\|\mathbf{x}\|_1 = 1$), the Markov chain operating on the graph has one unique stationary distribution vector $\boldsymbol{\pi}^*$ such that $\lim_{t \to \infty} \hat{\mathbf{A}}^t \mathbf{x} = \boldsymbol{\pi}^*$, where $\boldsymbol{\pi}^*(i) = \frac{d(v_i)}{2|\mathcal{E}|}$. If matrix $\hat{\mathbf{A}}$ is symmetric (i.e., $G$ is undirected), $\boldsymbol{\pi}^*$ will be a uniform distribution over the nodes, i.e., $\boldsymbol{\pi}^*(i) = \frac{1}{n}$.*

*Proof.* The stationary distribution vector existence and uniqueness has been proved in Norris (1998). Here, we need to identify on vector $\boldsymbol{\pi}$ at convergence such that $\hat{\mathbf{A}}\boldsymbol{\pi} = \boldsymbol{\pi}$, i.e.,

$$\boldsymbol{\pi}(i) = \sum_j \hat{\mathbf{A}}(i,j)\boldsymbol{\pi}(j). \tag{16}$$

According to the definition of $\hat{\mathbf{A}}$, it is easy to have

$$\sum_j \frac{w_{i,j}}{d_w(j)}\boldsymbol{\pi}(j) = \boldsymbol{\pi}(i). \tag{17}$$

where $w_{i,j}$ denotes the initial connection weight between $v_i$ and $v_j$. For the unweighted network, $w_{i,j}$ will be in $\{0, 1\}$ indicating if $v_i$ and $v_j$ are connected or not. Notation $d_w(i)$ denotes the rough degree of node $v_i$ in the network subject to the weight $w : \mathcal{E} \in \mathbb{R}$, which sums the weight of edges connected to the nodes in the network. So, it is enough to have $\boldsymbol{\pi}(j) \propto d_w(j)$. More precisely, we can set

$$\boldsymbol{\pi}(j) = \frac{d_w(j)}{\sum_k d_w(j)} = \frac{d_w(j)}{2|\mathcal{E}|}. \tag{18}$$

In this case,

$$\sum_j \hat{\mathbf{A}}(i,j)\boldsymbol{\pi}(j) = \sum_j \frac{w_{i,j}}{d_w(j)}\frac{d_w(j)}{2|\mathcal{E}|} = \sum_j \frac{w_{i,j}}{2|\mathcal{E}|} = \frac{d_w(i)}{2|\mathcal{E}|} = \boldsymbol{\pi}(i). \tag{19}$$

Meanwhile, for the symmetric and normalized adjacency matrix $\hat{\mathbf{A}}$, we can prove the stationary distribution $\boldsymbol{\pi}^*(i) = \frac{1}{n}$ in a similar way, which concludes the proof. $\square$

### 10.2.2 PROOF OF THEOREM 1

**THEOREM 1.** *Given a input network $G = (\mathcal{V}, \mathcal{E})$, which is unweighted, irreducible, finite and aperiodic, if there exist enough nested Markov chain layers in the GCN model, it will reduce the nodes' representations from the column-normalized feature matrix $\mathbf{X} \in \mathbb{R}^{n \times d_x}$ to the stationary representation $\mathbf{\Pi}^* = [\boldsymbol{\pi}^*, \boldsymbol{\pi}^*, \cdots, \boldsymbol{\pi}^*] \in \mathbb{R}^{n \times d_x}$. Furthermore, if $G$ is undirected, then the stationary representation will become $\mathbf{\Pi}^* = \frac{1}{n} \cdot \mathbf{1}^{n \times d_x}$.*

*Proof.* This theorem can be proved based on Lemma 1. For any initial state distribution vector $\mathbf{x} \in \mathbb{R}^{|\mathcal{V}| \times 1}$, for the Markov chain at convergence, we have

$$\lim_{t \to \infty} \hat{\mathbf{A}}^t \mathbf{x} = \boldsymbol{\pi}^*. \tag{20}$$

We misuse the notation $\hat{\mathbf{A}}^* = \lim_{t \to \infty} \hat{\mathbf{A}}^t$. In this case,

$$\begin{aligned}
\hat{\mathbf{A}}^*\mathbf{X} &= [\hat{\mathbf{A}}^*\mathbf{X}(:,1), \hat{\mathbf{A}}^*\mathbf{X}(:,2), \cdots, \hat{\mathbf{A}}^*\mathbf{X}(:,d_x)] \\
&= [\boldsymbol{\pi}^*, \boldsymbol{\pi}^*, \cdots, \boldsymbol{\pi}^*],
\end{aligned} \tag{21}$$

which together with Lemma 1 conclude the proof. $\square$

### 10.2.3 PROOF OF THEOREM 2

Prior to introducing the proof of Theorem 2, we will introduce the following lemma first.

**LEMMA 2.** *For any vector* $\mathbf{x} \in \mathbb{R}^n$, *the following inequality holds:*

$$\|\mathbf{x}\|_p \le (n)^{\frac{1}{p} - \frac{1}{q}} \|\mathbf{x}\|_q. \tag{22}$$

*Proof.* According to Hölder's inequality Hölder (1889), for $\forall \mathbf{a}, \mathbf{b} \in \mathbb{R}^{n \times 1}$ and $r > 1$,

$$\sum_{i=1}^n |\mathbf{a}(i)||\mathbf{b}(i)| \le \left( \sum_{i=1}^n |\mathbf{a}(i)|^r \right)^{\frac{1}{r}} \left( \sum_{i=1}^n |\mathbf{b}(i)|^{\frac{r}{r-1}} \right)^{1 - \frac{1}{r}}. \tag{23}$$

Let $|\mathbf{a}(i)| = |\mathbf{x}_i|^p$, $|\mathbf{b}(i)| = 1$ and $r = \frac{q}{p}$,

$$\begin{aligned}
\sum_{i=1}^n |\mathbf{x}_i|^p &= \sum_{i=1}^n |\mathbf{x}_i|^p \cdot 1 \\
&\le \left( \sum_{i=1}^n (|\mathbf{x}_i|^p)^{\frac{q}{p}} \right)^{\frac{p}{q}} \left( \sum_{i=1}^n 1^{\frac{q}{q-p}} \right)^{1 - \frac{p}{q}} \\
&= \left( \sum_{i=1}^n |\mathbf{x}_i|^q \right)^{\frac{p}{q}} (n)^{1 - \frac{p}{q}}.
\end{aligned} \tag{24}$$

Therefore,

$$\begin{aligned}
\|\mathbf{x}\|_p &= \left( \sum_{i=1}^n |\mathbf{x}(i)|^p \right)^{\frac{1}{p}} \\
&\le \left( \left( \sum_{i=1}^n |\mathbf{x}_i|^q \right)^{\frac{p}{q}} (n)^{1 - \frac{p}{q}} \right)^{\frac{1}{p}} \\
&= \left( \sum_{i=1}^n |\mathbf{x}_i|^q \right)^{\frac{1}{q}} (n)^{\frac{1}{p} - \frac{1}{q}} \\
&= (n)^{\frac{1}{p} - \frac{1}{q}} \|\mathbf{x}\|_q.
\end{aligned} \tag{25}$$

$\square$

**THEOREM 2.** *Let* $1 \ge \lambda_1 \ge \lambda_2 \ge \cdots \ge \lambda_n$ *be the eigen-values of matrix* $\hat{\mathbf{A}}$ *defined based on network* $G$, *then the corresponding suspended animation limit of the* GCN *model on* $G$ *is tightly bounded*

$$\zeta \le \mathcal{O} \left( \frac{\log \min_i \frac{1}{\boldsymbol{\pi}^*(i)}}{1 - \max\{\lambda_2, |\lambda_n|\}} \right). \tag{26}$$

*In the case that the network* $G$ *is a d-regular, then the suspended animation limit of the* GCN *model on* $G$ *can be simplified as*

$$\zeta \le \mathcal{O} \left( \frac{\log n}{1 - \max\{\lambda_2, |\lambda_n|\}} \right). \tag{27}$$

*Proof.* Instead of proving the above inequality directly, we propose to prove that $\zeta$ is *suspended animation limit* by the following inequality instead

$$\left\| \hat{\mathbf{A}}^\zeta \mathbf{x} - \boldsymbol{\pi}^* \right\|_2 \le \frac{\epsilon}{\sqrt{n}}, \tag{28}$$

which can derive the following inequality according to Lemma 2:

$$\left\| \hat{\mathbf{A}}^\zeta \mathbf{x} - \boldsymbol{\pi}^* \right\|_1 \le \epsilon. \tag{29}$$

Let $\mathbf{v}_1, \mathbf{v}_2, \cdots, \mathbf{v}_n$ be the eigenvectors of $\hat{\mathbf{A}}$ and $\forall \mathbf{x}$

$$\mathbf{x} = \sum_i \langle \mathbf{x}, \mathbf{v}_i \rangle \, \mathbf{v}_i = \sum_i \alpha_i \mathbf{v}_i, \tag{30}$$

where $\alpha_i = \langle \mathbf{x}, \mathbf{v}_i \rangle$.

Therefore, we have

$$\hat{\mathbf{A}}^\zeta \mathbf{x} = \sum_i \alpha_i \lambda_i^\zeta \mathbf{v}_i. \tag{31}$$

$\square$

Considering that $\lambda_1^\zeta = 1$ and $\mathbf{v}_1 = [\frac{1}{\sqrt{n}}, \frac{1}{\sqrt{n}}, \cdots, \frac{1}{\sqrt{n}}]^\top$, then

$$\alpha_1 = \langle \mathbf{x}, \mathbf{v}_1 \rangle = \sum_i \mathbf{x}(i) \frac{1}{\sqrt{n}} = \frac{1}{\sqrt{n}} \|\mathbf{x}\|_1 = \frac{1}{\sqrt{n}}, \tag{32}$$

and

$$\alpha_1 \lambda_1^\zeta \mathbf{v}_1 = \frac{1}{\sqrt{n}} \mathbf{v}_1 = [\frac{1}{n}, \frac{1}{n}, \cdots, \frac{1}{n}] = \boldsymbol{\pi}^*, \tag{33}$$

where

Therefore, we have

$$\begin{aligned}
\left\| \hat{\mathbf{A}}^\zeta \mathbf{x} - \boldsymbol{\pi}^* \right\|_2^2 &= \left\| \sum_{i=2} \alpha_i \lambda_i^\zeta \mathbf{v}_i \right\|_2^2 = \sum_{i=2} \alpha_i^2 \lambda_i^{2\zeta} \\
&\leq \sum_{i=2} \alpha_i^2 \lambda_{max}^{2\zeta} = \sum_{i=2} \alpha_i^2 \mathbf{v}_i^\top \mathbf{v}_i \lambda_{max}^{2\zeta} \\
&\leq \|\mathbf{x}\|_2^2 \lambda_{max}^{2\zeta} \\
&\leq \lambda_{max}^{2\zeta},
\end{aligned} \tag{34}$$

where $\lambda_{max} = \max\{|\lambda_2|, |\lambda_3|, \cdots, |\lambda_n|\} = \max\{\lambda_2, |\lambda_n|\}$.

Therefore, to ensure

$$\max\{\lambda_2, |\lambda_n|\} \leq \frac{\epsilon^2}{n} \tag{35}$$

we can have

$$\zeta \leq \frac{\log \epsilon - \log n}{2 \log \lambda_{max}} \leq \mathcal{O}\left( \frac{\log n}{1 - \lambda_{max}} \right). \tag{36}$$

### 10.2.4 Proof of Corollary 1

**COROLLARY 1.** *Let $1 \geq \lambda_1 \geq \lambda_2 \geq \cdots \geq \lambda_n$ be the eigen-values of matrix $\hat{\mathbf{A}}$ defined based on network $G$, then the corresponding suspended animation limit of the GCN model (with lazy Markov chain based layers) on $G$ is tightly bounded*

$$\zeta \leq \mathcal{O}\left( \frac{\log \min_i \frac{1}{\boldsymbol{\pi}^*(i)}}{1 - \lambda_2} \right). \tag{37}$$

*Proof.* For the *lazy Markov chain* layer, we have its updating equation as follows

$$\mathbf{T} = \frac{1}{2} \hat{\mathbf{A}} \mathbf{H}^{(k-1)} + \frac{1}{2} \mathbf{H}^{(k-1)} = \frac{1}{2}(\hat{\mathbf{A}} + \mathrm{diag}(\{d_w(i)\}_{v_i \in \mathcal{V}})) \mathbf{H}^{(k-1)} = \tilde{\mathbf{A}} \mathbf{H}^{(k-1)}. \tag{38}$$

It is easy to show that $\tilde{\mathbf{A}}$ is positive definite and we have its eigen-values $\lambda_1 \geq \lambda_2 \geq \cdots \geq \lambda_n \geq 0$. Therefore,

$$\lambda_{\max} = \max\{|\lambda_2|, |\lambda_3|, \cdots, |\lambda_n|\} = \max\{\lambda_2, |\lambda_n|\} = \lambda_2, \tag{39}$$

which together with Theorem 2 conclude the proof. $\square$

### 10.3 PROOF OF THEOREM 3

Prior to introducing the proof of Theorem 3, we first introduce the following lemma.

**LEMMA 3.** *For any non-singular matrix* $\mathbf{I} + \mathbf{M}$*, we have*

$$1 - \sigma_{max}(\mathbf{M}) \leq \sigma_{min}(\mathbf{I} + \mathbf{M}) \leq \sigma_{max}(\mathbf{I} + \mathbf{M}) \leq 1 + \sigma_{max}(\mathbf{M}), \tag{40}$$

*where* $\sigma_{max}(\cdot)$ *and* $\sigma_{min}(\cdot)$ *denote the maximum and minimum singular values of the input matrix, respectively.*

*Proof.* Due to the triangle inequality, the upper bound is easy to prove:

$$\sigma_{max}(\mathbf{I} + \mathbf{M}) = \|\mathbf{I} + \mathbf{M}\|_2 \leq \|\mathbf{I}\|_2 + \|\mathbf{M}\|_2 = 1 + \sigma_{max}(\mathbf{M}). \tag{41}$$

In the case that $\sigma_{max}(\mathbf{M}) \geq 1$, the lower bound is trivial to prove since $\mathbf{I} + \mathbf{M}$ is non-singular, we have

$$\sigma_{min}(\mathbf{I} + \mathbf{M}) > 0. \tag{42}$$

Meanwhile, in the case that $\sigma_{max}(\mathbf{M}) < 1$, it is easy to know that $|\lambda_{max}(\mathbf{M})| < 1$, where $\lambda_{max}(\cdot)$ denotes the latest eigenvalue of the input matrix.

$$
\begin{aligned}
\sigma_{min}(\mathbf{I} + \mathbf{M}) = \left\|(\mathbf{I} + \mathbf{M})^{-1}\right\|_2^{-1} &= \left\|\sum_{k=1}^{\infty}(-1)^k \mathbf{M}^k\right\|_2^{-1} \\
&\geq \left(\sum_{k=1}^{\infty}\left\|(-1)^k \mathbf{M}^k\right\|_2\right)^{-1} \\
&\geq \left(\sum_{k=1}^{\infty}\|\mathbf{M}\|_2^k\right)^{-1} \\
&= \left(\frac{1}{1 - \|\mathbf{M}\|_2}\right)^{-1} = 1 - \sigma_{max}(\mathbf{M}),
\end{aligned}
\tag{43}
$$

which concludes the proof for the lower bound. $\quad\square$

**THEOREM 3.** *Let $H$ denote the objective function that we want to model, which satisfies Assumption 1, in learning the $K$-layer GRESNET model, we have the following inequality hold:*

$$(1 - \delta)\left\|\frac{\partial \ell(\mathbf{\Theta})}{\partial \mathbf{x}^{(k)}}\right\|_2 \leq \left\|\frac{\partial \ell(\mathbf{\Theta})}{\partial \mathbf{x}^{(k-1)}}\right\|_2 \leq (1 + \delta)\left\|\frac{\partial \ell(\mathbf{\Theta})}{\partial \mathbf{x}^{(k)}}\right\|_2, \tag{44}$$

*where* $\delta \leq c\frac{\log(2K)}{K}$ *and* $c = c_1 \max\{\alpha\beta(1 + \beta), \beta(2 + \alpha) + \alpha\}$ *for some* $c_1 > 0$.

*Proof.* We can represent the Jacobian matrix $\mathbf{J}$ of $\mathbf{x}^{(k)}$ with $\mathbf{x}^{(k-1)}$. Therefore, we have

$$\frac{\partial \ell(\mathbf{\Theta})}{\partial \mathbf{x}^{(k-1)}} = \frac{\partial \ell(\mathbf{\Theta})}{\partial \mathbf{x}^{(k)}}\frac{\partial \mathbf{x}^{(k)}}{\partial \mathbf{x}^{(k-1)}} = \mathbf{J}\frac{\partial \ell(\mathbf{\Theta})}{\partial \mathbf{x}^{(k)}}. \tag{45}$$

Matrix $\mathbf{J}$ can be rewritten as $\mathbf{J} = \mathbf{I} + \nabla F^{(k-1)}(\mathbf{x}^{(k-1)})$, where

$$\nabla F^{(k-1)}(\mathbf{x}^{(k-1)}) = \lim_{t \to 0^+}\left\|\frac{F^{(k-1)}(\mathbf{x}^{(k-1)} + t\mathbf{v}) - F^{(k-1)}(\mathbf{x}^{(k-1)})}{t}\right\|_2. \tag{46}$$

Meanwhile, it is easy to know that

$$\sigma_{min}(\mathbf{J})\left\|\frac{\partial \ell(\mathbf{\Theta})}{\partial \mathbf{x}^{(k)}}\right\|_2 \leq \left\|\mathbf{J}\frac{\partial \ell(\mathbf{\Theta})}{\partial \mathbf{x}^{(k)}}\right\|_2 \leq \sigma_{max}(\mathbf{J})\left\|\frac{\partial \ell(\mathbf{\Theta})}{\partial \mathbf{x}^{(k)}}\right\|_2. \tag{47}$$

$\quad\square$

Based on the above lemma, we have

$$(1 - \sigma) \left\| \frac{\partial \ell(\boldsymbol{\Theta})}{\partial \mathbf{x}^{(k)}} \right\|_2 \leq \left\| \frac{\partial \ell(\boldsymbol{\Theta})}{\partial \mathbf{x}^{(k-1)}} \right\|_2 \leq (1 + \sigma) \left\| \frac{\partial \ell(\boldsymbol{\Theta})}{\partial \mathbf{x}^{(k)}} \right\|_2, \tag{48}$$

where $\sigma = \sigma_{max}(\nabla F^{(k-1)}(\mathbf{x}^{(k-1)}))$.

Furthermore, we know that

$$\begin{aligned} \sigma_{max}(\nabla F^{(k-1)}(\mathbf{x}^{(k-1)}) &= sup_{\mathbf{v}} \frac{\left\| \nabla F^{(k-1)}(\mathbf{x}^{(k-1)})\mathbf{v} \right\|_2}{\|\mathbf{v}\|_2} \\ &= \lim_{t \to 0^+} sup_{\mathbf{v}} \frac{\left\| F^{(k-1)}(\mathbf{x}^{(k-1)} + t\mathbf{v}) - F^{(k-1)}(\mathbf{x}^{(k-1)}) \right\|_2}{t \|\mathbf{v}\|_2} \\ &\leq \left\| F^{(k-1)} \right\|_L, \end{aligned} \tag{49}$$

where $\|\cdot\|_L$ denotes the Lipschitz seminorm of the input function and it is defined as

$$\left\| F^{(k-1)} \right\|_L = sup_{\mathbf{x} \neq \mathbf{y}} \frac{\left\| F^{(k-1)}(\mathbf{x}) - F^{(k-1)}(\mathbf{y}) \right\|_2}{\|\mathbf{x} - \mathbf{y}\|_2}. \tag{50}$$

Meanwhile, according to the Theorem 1 in Bartlett et al. (2018) (whose proof will not be introduced here), we know that

$$\left\| F^{(k-1)} \right\|_L \leq c \frac{\log 2K}{K}, \tag{51}$$

which concludes the proof. In the above equation, $K$ denotes the layer depth of the model.

