# OpenReview forum: "GResNet: Graph Residual Network for Reviving Deep GNNs from Suspended Animation"
_ICLR.cc/2020/Conference — Reject_

### Official Review · AnonReviewer3 · 2019-10-14
**Official Blind Review #3**

**Rating:** 3

**Review:**

In this paper, the authors study the problem of adding residual connection to GNN for node classification. The authors first identify the problem referred to as “Suspended Animation” in GNN when the depth increases. Then the authors provide both empirical and theoretical characterization for the behavior. To handle this issue, the authors study several different ways to add residual connections in GNN including the naïve method proposed in Kipf et al. The authors carry out extensive experiments on three datasets on different residual connections for the node classification task.

Strength
1. The authors identify and study an interesting and important issue in GNN as the “Suspended Animation” issue.
2. The authors provide both empirical and theoretical analysis for the “Suspended Animation” behavior. Moreover, the authors provide theoretical justification for the added residual connection in GNN as gradient norm bound.
3. The authors carry out extensive experiments on different variants of residual links. Morover, the authors provide code online for reproducibility.

Weakness:
1. In the theoretical analysis, the assumption that the FC layer is identical mapping is too simplistic. The analysis differs from the actual model especially when the residual links are considered in equation (8), where we have a sum of FC layer output and residual connection. Actually, the empirical results show that naïve residual links work pretty good on several datasets. It goes against the analysis in Corollary 1.
2. Though the authors provide bound on the norm of gradient under residual links, it would be better if authors could justify the adding of residual link from the perspective of Theorem 1.
3. The authors study the depth of GNN up to 7 layers at most. It would be interesting to see how the model performs under really deep networks.
4. The authors mention several factors to affect GNN in section 4.2. It would be interesting to see how these factors like training data set and feature coding interacts with different ways of adding residual connections.
5. It is very informative that the authors compare their methods on the widely used three datasets. It would be better if the authors could carry out experiment on larger graphs to verify the empirical observations. For example, do we need to have deeper networks for larger graphs?


**Experience Assessment:**

I have published one or two papers in this area.

**Review Assessment: Checking Correctness Of Derivations And Theory:**

I assessed the sensibility of the derivations and theory.

**Review Assessment: Checking Correctness Of Experiments:**

I assessed the sensibility of the experiments.

**Review Assessment: Thoroughness In Paper Reading:**

I read the paper at least twice and used my best judgement in assessing the paper.

---

> ### Author Response · Authors · 2019-11-15
> **Response to Review #3 (Part 3, continued ...)**
>
>
> ******************
> Question 6: Experiments on larger graph datasets.
>
> Response: We clarify that Cora, Citeseer and Pubmed are the top three commonly used benchmark datasets in graph neural network studies, e.g., [1][2][3][4][5][6][7][8] etc. For comparison fairness with the existing works, we only show the experimental results on these three datasets in this paper.
>
> We do have also witnessed several other datasets used in the recent graph neural network papers, but majority of them are still the variants of Cora, Citeseer and Pubmed (by graph sampling or including more labeled data instances). The remaining ones are either private or rarely used by the existing papers. The reviewers can also refer to the webpage for more information about the major datasets studied by the community on graph neural network studies.
>
> https://paperswithcode.com/task/node-classification
>
> During the rebuttal period, as suggested by the reviewer, we also tried to get the experimental results of GResNet on the PPI and Reddit datasets as well, which are other public benchmark dataset used in graph neural network works [2][8]. The experimental results are reported in the updated version of the submitted PDF paper. The reviewer can refer to updated Table 4 in the appendix of the paper (on page 15) more information about the experimental results.
>
> We need to mention that GAT used in GResNet is very slow on large datasets (it is not our problem but the problem with the GAT base model), we cannot get the results of GResNet(GAT) out on Reddit during the rebuttal period. Since PPI and Reddit are not common used in other graph neural network papers, so many of the entries in Table 4 are not provided. We will add these results in the camera-ready version of this paper later.
>
> We will seek for more public graph benchmark datasets as suggested by the reviewer. It may take longer time than the rebuttal time allows, so we plan to add the comparison experiments in the camera-ready version of this paper instead.
>
> ******************
>
> Hope our response has resolved your concerns. If there is any proposed question about this paper not resolved in our response, welcome to let us know and we are happy to discuss more with you.
>
>
> References used in the above response:
> [1] GCN: Semi-Supervised Classification with Graph Convolutional Networks
> [2] GAT: Graph Attention Networks
> [3] DGI: Deep Graph Infomax
> [4] GraphStar: Graph Star Net for Generalized Multi-Task Learning
> [5] APPNP: Predict then Propagate: Graph Neural Networks meet Personalized PageRank
> [6] GOCN: Graph Optimized Convolutional Networks
> [7] GraphNAS: GraphNAS: Graph Neural Architecture Search with Reinforcement Learning
> [8] FastGCN: Fast Learning with Graph Convolutional Networks via Importance Sampling
> [9] Improving Random Walk Estimation Accuracy with Uniform Restart.

---

> ### Author Response · Authors · 2019-11-15
> **Response to Review #3 (Part 2, continued ...)**
>
>
>
> ******************
> Question 3: Justify the adding of residual links from the perspective of Theorem 1.
>
> Response: We clarify that the Corollary 1 is the justification for the naïve residual term actually, we can also drive similar bound terms for the raw residual terms based on Corollary 1 and [9]. For the remaining two residual terms, the derivation and proof will be very similar as well. We don't plan to study the residual term analysis in this way, since it will make the analysis sections too redundant and also fragmented, and also introduce multiple Corollaries with duplicated contents/words.
>
> If the reviewer can further take a look at Theorem 3, we clarify that Theorem 3 used in the paper is more appropriate for illustrating the effectiveness of the proposed residual terms due to:
> (1) except the assumption of the mapping properties, it has no other assumptions on the model.
> (2) it works for all these residual terms, and we don’t need to case by case analysis.
> (3) The gap term $\delta = c \cdot \frac{\log 2K}{K}$ illustrates that deeper model with our proposed residual terms tend to lead to smaller gaps instead.
>
> If the reviewer really think the case by case study on the residual terms is better, we will also add it to the appendix as another way to illustrate the effectiveness of the residual terms. However, we still have the concerns, since it will make the analysis section looks long-winded and also fragmented, and it is also duplicated with the current Theorem 3.
>
> Hope the reviewer can understand our concern.
>
> ******************
>
> Question 4: The authors study the GNNs up to 7 layers at most.
>
> Response: We clarify that we focus on illustrating the suspended animation problem with the existing GNNs, and we also intend to propose a feasible solution to resolve the problem. For GCN, its suspended animation limit is 5, so we only show the experiment results of the model with depth up to 7 in the original submission.
>
> Actually, we have studied GCN to 50 layers. We have added the extra experimental results on GCN with very deep layers in Figure 4-8 in the appendix (Sec 8.2 on page 16-17 of the updated PDF submission of the paper), just for the reviewer’s information. According to the results, we can observe that GCN, GResNet(naive) and GResNet(graph-naive) will all fail to work as the model depth increases to 20 and more. However, GResNet(raw) and GResNet(graph-raw) can still work very well for very deep architectures.
>
> ******************
>
> Question 5: Impact of the other factors together with the residual terms on the performance of the models.
>
> Response: We clarify that we have the analysis results available on these factors actually. However, since the results are not very closely related to the analysis of suspended animation problem , so we didn’t put them in the paper.
>
> Since the reviewer point this out, we also add the results in the paper appendix to illustrate the impact of the other factors together with the residual terms on the models. The results are available in Section 9 in the appendix (on page 18) of the updated PDF submission above.

---

> ### Author Response · Authors · 2019-11-15
> **Response to Review #3 (Part 1)**
>
>
>
> First of all, we would like to appreciate the reviewer for the constructive suggestions. There is a space limit for our response textual input, so we will split our response into three parts. Here is the part 1, and part 2 and part 3 are posted below. The mentioned references are at the very bottom of the response.
>
> ******************
> Question 1: Assumption on the identical assumption for the FC layer.
>
> Response: Firstly, we agree with the reviewer that we may need to involve the non-linear layers into the model performance analysis if we can, since the non-linear layers tend to affect the suspended animation problem as well. The impacts come from multiple perspectives, one of the impact is caused by the gradient vanishing/exploding problem in learning the model with multiple non-linear layers. Also the non-linear layers will project the node representations to random states other than the stationary representations.
>
> However, we also hope the reviewer can understand the reason that we make that assumption. As we all know, analysis with the multiple non-linear layers is a difficult task, the community still have great challenges in interpreting and explaining the learning process of deep learning models by this context so far. Therefore, we propose to simplify the analysis settings based on the assumption mentioned in the reviewer's comments. We would like to clarify that our assumption also makes sense actually in certain perspectives illustrated as follows.
>
> (1) Non-linear layer will not revive the suspended animated models. We clarify that if a model suffers from the suspended animation problem, the non-linear layer will not change the suspended limitation problem. As clarified in the second paragraph of Sec 3.2 on page 4, the parameter W is actually shared for all the nodes in the GCN model. For any two nodes which obtain the identical representations (i.e., suffering from the suspended animation happens as stated in Theorem 1), their representations obtained via the non-linear layer will still be the same (i.e., the projected representations will still have the suspended animation problem).
>
> (2) Analysis feasibility. Involving the non-linear layers in the model analysis will make the graph neural network model performance and the animation limit bound introduced in Sec 4 extremely hard (or even impossible) to study. It will still due to the black-box property of the deep models with multiple non-linear projections.
>
> (3) Markov-Chain layer is our main focus. We also have experimental studies on node classification with the same dataset by using MLP as the model, which will not suffer from the suspended animation problem at all. Meanwhile, compared with MLP, the main differences between GCN and MLP is due to the Markov-Chain layers involved in the model. Therefore, in this paper, we propose to be focused on studying the impact of the Markov-Chain layer on the model learning performance.
>
>
> ******************
>
> Question 2: Naïve residual links work pretty good on the several datasets.
>
> Response: We clarify that to check if naïve residual links can work or not, we may need to check Figure3(a) and Figure3(b). From the plots, we observe that for the GResNet(GCN,naïve) with less than 4 layers, we also acknowledge that its learning performance is not bad. However, from Figure 1, we observe that GCN without the naïve residual term can already perform very good with no greater than 5 layers. In other words, the good performance for GResNet(GCN,naïve) with no greater than 5 layers is not due to the naïve residual terms, since the base model is not bad for these cases.
>
> However, we it comes to the GResNet(GCN,naïve) model with 5,6,7 layers. From Figure3(a) and Figure3(b), we cannot really say that “naïve residual links work pretty good on several datasets” to resolve the suspended animation problem, since they don't really work well actually. If we further check Figure 3(e)-3(h), the curve differences between GResNet(GCN,naïve) vs GResNet(GCN,graph-raw) is very clear. I think the reviewer can also agree on this.
>
> To further help resolve the reviewer's concern on this point, we also add new experimental results on the GCN and GResNet models with deeper architectures on the Cora dataset, where the model depth include 2, 10, 20, 30, 40, and 50. The results are reported in Section 8.2 on pages 16-17 in the appendix of the updated PDF submitted above.
>
> According to the new Table 5 and Figures 4-8, we can observe that the naive, graph-naive residual terms will fail to work as the model depth further increases to 10, 20, and even more. However, raw and graph-raw residual terms can still work very well for the models even with 50 layers. Hope the new results will help resolve your question.

---

### Official Review · AnonReviewer1 · 2019-10-21
**Official Blind Review #1**

**Rating:** 3

**Review:**

Summary:

This paper studies the “suspended animation limit” of various GNNs – an important one for how to train a good Graph network. The authors provide sufficient analysis by simplify GNNs as a series of 1-step Markov chains (which is my concern as stated in the section on main issues), while pointing out the limitations quantitatively as in the Theorem 2. Under the assumption, the authors propose several new forms of ResNets for GCNs, which can successfully overcome the limitation.

Overall, the motivation of this work is clear and meaninfgful. The proposed residual architecture is effective, and the presentation is clear and easy to understand.

However, my main concerns are on the initial assumptions for analyzing the suspension of GNNs. See the following comments.

This paper is generally well written and easy to understand. The organization of each part is well-balanced.

Originality:

To the best of my knowledge, numerous methods (i.e., targeting on applications) address this problem by augmenting A [1] or X [2] with similarity of feature representation learned from other sources. However, this paper specifically analyzes the problem in a principle way. I consider this work is generally novel.

[1] X. Wang and A. Gupta. Videos as Space-Time Region Graphs. ECCV 2018.
[2] N. Wang, Y. Zhang, Z. Li , Y. Fu, W, Liu, Y. Jiang. Pixel2Mesh: Generating 3D Mesh Models from Single RGB Images. ECCV 2018.

Significance:

The significance lies mostly in motivation and the proposed GResNet.

Motivation: This paper studies the phenomenal that GNNs tends to not respond to the input data when certain depth of a network is reached, which the authors called as suspended animation limit. Studying problem is fundamental and important, and also unique since different with CNNs where data are independent, the data instance within GNNs are highly correlated.

GResNet: Given the differences, and within the Theorem 2 where the residual formulation of CNNs does not apply to GNNs, the paper also proposes several new formulations, i.e., in figure 2, where the suspension is avoidable and the performance under the same experimental settings is obviously boosted.

Main issues:
My major concern to this work lies in the assumption used throughout section 3 and 4. At the beginning of Section 3.2, the authors assume that W is identity. Since X is assumed to be column-wise normalized, the nonlinearity is removable. However, this is not true in real cases: W is actually learnable and not bounded. When W is learned to be negative, Relu layer is not removable, and the behavior of the network will be completely different with what the paper depicted. Indeed, GNNs contain stacked linear+nonlinear functions, which cannot be simplified as a linear Markov chain. It is analogy to CNNs, which is not possibly be simplified as a group of average poolings.

Minor issues:
1. I agree that under the assumption, eq. (11) shows that the differences between the learned representations are not discriminative, however the claim “majority of the nodes are of very small degrees” is not justified and only apply to the internet topology in Faloutsos et al. (1999).

2. I feel the “Raw Feature Coding” and the “Network Degree Distribution” are sort of repetitive, and the eq. (11) is eq. (12) at the stationary point.


**Experience Assessment:**

I have published in this field for several years.

**Review Assessment: Checking Correctness Of Derivations And Theory:**

I assessed the sensibility of the derivations and theory.

**Review Assessment: Checking Correctness Of Experiments:**

I assessed the sensibility of the experiments.

**Review Assessment: Thoroughness In Paper Reading:**

I read the paper at least twice and used my best judgement in assessing the paper.

---

> ### Author Response · Authors · 2019-11-15
> **Response to Review #1 (Part 2, Continued...)**
>
>
>
> ******************
> Question 3: “majority of the nodes are of very small degrees” is not justified and only apply to the internet topology in Faloutsos et al. (1999)
>
> Response: We clarify that node degree “power-law distribution” is well-known concept in graph studies. It depicts the observation that “majority of the nodes in a graph will have a small degree, and a small amount of the nodes can have a large degree”. If the reviewer has the Faloutsos et al. (1999) paper available, it is suggested to refer to the Figure 5 and Figure 6 in the paper. These two plots are the representative plots on node degree power-law distribution. It is a log-log plot on node degree (the x axis) and the node number (the y axis). From the plot we can observe that majority of the nodes have a degree less than 10 actually.
>
> Similar observation has been reported on the bibliographic network data as well in [3] (Cora, Citeseer and Pubmed datasets used in this paper are all bibliographic networks actually). The reviewer can refer to Figure 6 in [3] for more information on the related bibliographic networks. We have also cited this related paper in our updated paper submitted above, and also added necessary words to make this concept clearer in Sec 4.2.
>
> [3] M. E. J. Newman. The structure and function of complex networks
>
> ******************
> Question 4: I feel the “Raw Feature Coding” and the “Network Degree Distribution” are sort of repetitive, and the eq. (11) is eq. (12) at the stationary point.
>
> Response: We clarify that “Raw Feature Coding” and “Network Degree Distribution” are not repetitive and they are totally different factors. Their analyses are also very different actually.
>
> We want to clarify that Equ (11) used for the “Network Degree Distribution” is at the stationary point. However, Equ (12) for “Raw Feature Coding” is based on the raw representation in Equ (3) and Equ (4), which doesn’t require the representations to be at the stationary point.
>
> As suggested by the reviewer, we have also revised and updated the presentations for these two factors in the updated version of this paper just to avoid unnecessary confusions for the readers. The changes are added just after Equ(12) in the updated PDF of this paper.
>
> ******************
>
> Hope our response has resolved your concerns. If there is any proposed question about this paper not resolved in our response, welcome to let us know and we are happy to discuss more with you.

---

> ### Author Response · Authors · 2019-11-15
> **Response to Review #1 (Part 1)**
>
> First of all, we would like to appreciate the reviewer for the constructive suggestions. There is a space limit for our response textual input, so we will split our response into two parts. Here is the part 1, and part 2 will be posted as follows.
>
> ******************
> Question 1: Concerns with the assumption in Section 3 and 4.
>
> Response: Firstly, we agree with the reviewer that we may need to involve the non-linear layers into the model performance analysis if we can, since the non-linear layers tend to affect the suspended animation problem as well. The impacts come from multiple perspectives, one of the impact is caused by the gradient vanishing/exploding problem in learning the model with multiple non-linear layers. Also the non-linear layers will project the node representations to random states other than the stationary representations.
>
> However, we also hope the reviewer can understand the reason that we make that assumption. As we all know, analysis with the multiple non-linear layers is a difficult task, the community still have great challenges in interpreting and explaining the learning process of deep learning models by this context so far. Therefore, we propose to simplify the analysis settings based on the assumption mentioned in the reviewer's comments. We would like to clarify that our assumption also makes sense actually in certain perspectives illustrated as follows.
>
> (1) Non-linear layer will not revive the suspended animated models. We clarify that if a model suffers from the suspended animation problem, the non-linear layer will not change the suspended limitation problem. As clarified in the second paragraph of Sec 3.2 on page 4, the parameter W is actually shared for all the nodes in the GCN model. For any two nodes which obtain the identical representations (i.e., suffering from the suspended animation happens as stated in Theorem 1), their representations obtained via the non-linear layer will still be the same (i.e., the projected representations will still have the suspended animation problem).
>
> (2) Analysis feasibility. Involving the non-linear layers in the model analysis will make the graph neural network model performance and the animation limit bound introduced in Sec 4 extremely hard (or even impossible) to study. It will still due to the black-box property of the deep models with multiple non-linear projections.
>
> (3) Markov-Chain layer is our main focus. We also have experimental studies on node classification with the same dataset by using MLP as the model, which will not suffer from the suspended animation problem at all. Meanwhile, compared with MLP, the main differences between GCN and MLP is due to the Markov-Chain layers involved in the model. Therefore, in this paper, we propose to be focused on studying the impact of the Markov-Chain layer on the model learning performance.
>
> ******************
>
> Question 2: Originality and existing related works.
>
> Response: We clarify that the contributions of this paper involve four main parts: (1) identify a problem with existing graph neural network models; (2) study the causes and explain the problem; and (3) propose a tentative solution to resolve the problem; and (4) analyze the feasibility of the proposed solution.
>
> We just check the referred two papers mentioned by the reviewer, and we agree that the methods proposed in these two papers are very close to the solution proposed in our paper. We clarify that the main contributions of this paper lie in all these four parts mentioned above actually, not just the proposed solution (i.e., part (3) on the tentative solution).
>
> We also appreciate the reviewer for pointing out the relevant literatures [1][2], and we have cited them properly in the updated PDF uploaded above (these two papers are mentioned on page 8, the paragraph below Figure 3 in Sec 5.2).
>
> [1] X. Wang and A. Gupta. Videos as Space-Time Region Graphs. ECCV 2018.
> [2] N. Wang, Y. Zhang, Z. Li , Y. Fu, W, Liu, Y. Jiang. Pixel2Mesh: Generating 3D Mesh Models from Single RGB Images. ECCV 2018.

---

### Official Review · AnonReviewer2 · 2019-10-25
**Official Blind Review #2**

**Rating:** 6

**Review:**

The paper studies the causes of the empirically poor performance in deep structures that plagues existing GNNs, and identify the suspended animation problem as the main issue. In analogy to the Residual CNN network, a residual graph network is proposed to address such issue. Moreover, the underlying Markov chain property such as stationary distribution is theoretically analyzed, the so-called suspended animation limit is defined and its upper and lower bounds are established. Empirical experiments are relatived short and less sufficient, with comparisons on there datasets: Cora, Citeseer, and Pubmed. It would be more convincing to present its performance on a more diverse range of datasets.  Note the results on Citeseer is inferior to existing method. It is helpful to clearly explain why  this could be the case.

Post rebuttal edition: After reading the reviews and the authors' reply, several questions such as the major concerns over this oversimplified linear assumptions surface out, as discussed in length by other reviewers. Meanwhile, I still believe there are useful merits of this paper. Thus I adjust my current rating to weak accept.


**Experience Assessment:**

I have read many papers in this area.

**Review Assessment: Checking Correctness Of Derivations And Theory:**

I assessed the sensibility of the derivations and theory.

**Review Assessment: Checking Correctness Of Experiments:**

I assessed the sensibility of the experiments.

**Review Assessment: Thoroughness In Paper Reading:**

I read the paper at least twice and used my best judgement in assessing the paper.

---

> ### Author Response · Authors · 2019-11-15
> **Response to Reviewer #2**
>
> First of all, we would like to appreciate the reviewer for the support and the constructive suggestions.
>
> ******************
> Question 1: Empirical experiments on more diverse datasets other than Cora, Citeseer and Pubmed.
>
> We clarify that Cora, Citeseer and Pubmed are the top three commonly used benchmark datasets in graph neural network studies, e.g., [1][2][3][4][5][6][7][8] etc. For comparison fairness with the existing works, we only show the experimental results on these three datasets in this paper.
>
> We do have also witnessed several other datasets used in the recent graph neural network papers, but majority of them are still the variants of Cora, Citeseer and Pubmed (by graph sampling or including more labeled data instances). The remaining ones are either private or rarely used by the existing papers. The reviewers can also refer to the webpage for more information about the major datasets studied by the community on graph neural network studies.
>
> https://paperswithcode.com/task/node-classification
>
> During the rebuttal period, we tried to get the experimental results of GResNet on the PPI and Reddit datasets as well, which are two other public benchmark dataset used in some graph neural network works [2][8]. The experimental results are reported in the updated version of the submitted PDF paper. The reviewer can refer to Table 4 in the appendix (on page 15) of the updated PDF submitted above for more information about the experimental results.
>
> We need to mention that GAT used in GResNet is very slow on large datasets (it is not our problem but the problem with the GAT base model), we cannot get the results of GResNet(GAT) out on Reddit during the rebuttal period. Also since PPI and Reddit are not common used in other graph neural network papers, many of the entries in Table 4 are not provided. We will add these results in the camera-ready version of this paper later.
>
> We will seek for more public graph benchmark datasets as suggested by the reviewer. It may take longer time than the rebuttal time allows, so we plan to add the comparison experiments in the camera-ready version of this paper as well.
>
> ******************
>
> Question 2: Inferior experimental results on the Citeseer compared with the existing works.
>
> We clarify that the GResNet model proposed in this paper differs a lot from the latest graph neural network models, i.e., APPNP, GOCN as reported in Table 3. Instead of introducing new and complex model architectures (e.g., APPNP) or complicated optimization approaches (e.g., GOCN) merely for learning performance improvement, we aim to introduce a general framework, which can work for any base models to revive them from the suspended animation problem instead. So, the objective differences may lead to slightly different experimental performance of the models.
>
> As pointed out by the reviewer, the learning performance of our model GResNet(GCN) and GResNet(LoopyNet) is slightly inferior to APPNP and GOCN. Besides the main objective differences mentioned above for these different works, it can also be caused by:
> (1) weak base models GCN and LoopyNet used in these two methods, since we observe that GResNet(GAT) can achieve the best performance among all the comparison methods (also better than APPNP and GOCN);
> (2) slightly different learning settings, APPNP allows the model to involve more labeled training data for model learning, which may lead to slightly higher scores. However, we strictly follow the conventional learning setting (labeled data ratio) for our GResNet methods.
>
> The experimental results provided in Table 3 is just to provide the latest research results obtained by the current papers just for the reviewers' and readers' information. So, the slightly inferior performance of GResNet than APPNP doesn't necessarily indicate that GResNet is not good. The main objective of this paper is still focused on studying the suspended animation problem with deep graph neural network models, not to compare the learning scores.
>
> ******************
>
> Hope our response has resolved your concerns. If there is any proposed question about this paper not resolved in our response, welcome to let us know and we are happy to discuss more with you.
>
>
> References used in the above response:
> [1] GCN: Semi-Supervised Classification with Graph Convolutional Networks
> [2] GAT: Graph Attention Networks
> [3] DGI: Deep Graph Infomax
> [4] GraphStar: Graph Star Net for Generalized Multi-Task Learning
> [5] APPNP: Predict then Propagate: Graph Neural Networks meet Personalized PageRank
> [6] GOCN: Graph Optimized Convolutional Networks
> [7] GraphNAS: GraphNAS: Graph Neural Architecture Search with Reinforcement Learning
> [8] FastGCN: Fast Learning with Graph Convolutional Networks via Importance Sampling

---

### Author Response · Authors · 2019-11-15
**To All Reviewers (changes in the PDF submission)**

Hi All,

We have worked no the rebuttal for this submission in the past week, we also appreciate your constructive suggestions and comments. Hope our response can resolve your concerns about this paper. Some of the proposed concerns are hard (or impossible) to resolve actually, like analysis with the non-linear layers. It is not only hard for us in this paper, but also a very challenging task for the whole community to study the model performance bounds with such Multiple Non-linear layers. Really hope the reviewers can understand that.

Together with the responses posted below, we also updated the PDF submission by updating/adding/deleting some contents in the paper. The above PDF submission is slightly longer than 10 pages now and the reference is also longer than 2 pages. We will shrink the paper to the required page limit in the camera-ready version instead.


The main changes to the PDF submission include:

(1) New clarification to some concepts (e.g., power-law) are added. New reference papers are added.

(2) New experimental results added.

(2-1) In the new Section 8.1 and Table 4 in the appendix, we provide the results of GResNet on two other large-sized graph datasets, PPI and Reddit. These two datasets are not the common choices for graph neural network evaluation actually. Therefore, many entries of the baseline methods are not provided in Table 4. Also GAT is too slow, and not runnable on the large sized graph dataset (it is not our problem), so we cannot get the result of GResNet(GAT) out neither in one week. We will complement the Table 4 in the camera-ready version of this paper instead.

(2-2) In the new Section 8.2 (Table 5, and Figure 4-8) in the appendix, we provide the experimental results of the models with even deeper architectures, e.g., 10, 20, 30, 40, 50. According to the results, some of the residual terms (e.g., naive, graph-naive) will fail to work as the model goes deeper, but raw and graph raw residual terms can still work very well.

(2-3) In Section 9 (Table 6-7) in the appendix, we add the analysis and results about several other factors, including training set size and the raw feature encoding methods. Just for the reviewer's information.

---

### Decision · Program_Chairs · 2019-12-19

**Decision:**

Reject

**Comment:**

This paper studies the “suspended animation limit” of various graph neural networks (GNNs) and provides some theoretical analysis to explain its cause. To overcome the limitation, the authors propose Graph Residual Network (GRESNET) framework to involve nodes’ raw features or intermediate representations throughout the graph for all the model layers. The main concern of the reviewers is: the assumption made for theoretical analysis that the fully connected layer is identical mapping is too stringent. The paper does not gather sufficient support from the reviewers to merit acceptance, even after author response and reviewer discussion.  I thus recommend reject.